# Generalization Bounds for Stochastic Gradient Descent via Localized $\varepsilon$-Covers

**Sejun Park**
Korea University
`sejun.park000@gmail.com`

**Umut Şimşekli**
DI ENS, Ecole Normale Supérieure, Université PSL, CNRS, INRIA
`umut.simsekli@inria.fr`

**Murat A. Erdogdu**
University of Toronto & Vector Institute
`erdogdu@cs.toronto.edu`

## Abstract

In this paper, we propose a new covering technique localized for the trajectories of SGD. This localization provides an algorithm-specific complexity measured by the covering number, which can have dimension-independent cardinality in contrast to standard uniform covering arguments that result in exponential dimension dependency. Based on this localized construction, we show that if the objective function is a finite perturbation of a piecewise strongly convex and smooth function with $P$ pieces, i.e. non-convex and non-smooth in general, the generalization error can be upper bounded by $O(\sqrt{(\log n \log(nP))/n})$, where $n$ is the number of data samples. In particular, this rate is independent of dimension and does not require early stopping and decaying step size. Finally, we employ these results in various contexts and derive generalization bounds for multi-index linear models, multi-class support vector machines, and $K$-means clustering for both hard and soft label setups, improving the known state-of-the-art rates.

## 1 Introduction

We consider the following stochastic optimization problem

$$\min_{\theta \in \Theta} \left\{ F(\theta) := \mathbb{E}_Z[f(\theta; Z)] \right\}, \tag{1.1}$$

where $\theta$ represents the optimization parameter, $\Theta \subset \mathbb{R}^d$ is a convex parameter domain, $f(\,\cdot\,; z)$ is a possibly non-convex loss incurred by a single data point $z \in \mathcal{Z}$, and $Z$ is a random variable on $\mathcal{Z}$ following the data distribution. Since the distribution of $Z$ is unknown in general, the following proxy based on independent and identically distributed (i.i.d.) samples $z_1, \ldots, z_n$ of $Z$ is optimized instead

$$\min_{\theta \in \Theta} \left\{ \hat{F}(\theta) := \frac{1}{n} \sum_{i=1}^n f(\theta; z_i) \right\}. \tag{1.2}$$

Given a learning algorithm $A(\,\cdot\,)$ mapping samples $z_1, \ldots, z_n$ to an approximate solution of (1.2), bounding the *generalization error*[1] $\hat{F}(A(z_1, \ldots, z_n)) - F(A(z_1, \ldots, z_n))$ is a fundamental problem

---

[1]This quantity is also referred to as the generalization gap or the estimation error in the literature.

36th Conference on Neural Information Processing Systems (NeurIPS 2022).

in learning theory. Classical *algorithm-independent* results rely on uniform convergence over the entire domain $\Theta \subset \mathbb{R}^d$; thus, they apply to any learning algorithm. However, these bounds often increase with the dimension $d$ [SSSSS09, SSBD14, Fel16], becoming vacuous in the modern over-parameterized regime [ZBH+21]. To derive dimension-independent bounds, researchers have been investigating *algorithm-dependent* generalization properties, especially for commonly used training methods such as stochastic gradient descent (SGD) [HRS15, SHN+18, YKM21].

Notably, *algorithmic stability* is a technique for deriving generalization bounds based on the properties of a specific learning algorithm, which leverages that if a parameter learned by an algorithm is robust under a perturbation of samples $z_1, \ldots, z_n$, then the generalization error at that parameter must be small [BE02]. Based on this principle, several works proposed dimension-independent generalization bounds for SGD and its variants under various setups [HRS15, Lon17, FV19, LLQ20, BFGT20, LY20, FO21, LLY21, KWS22]. Bounds derived using algorithmic stability is optimal for strongly convex and smooth functions [SSSSS09] and convex and non-smooth functions [BFGT20, AKL21]. Nevertheless, without global (strong) convexity, early stopping, and/or decaying step size, generalization bounds based on algorithmic stability often diverge with the number of SGD iterations [HRS15, LLQ20], failing to explain the empirical observations.

To obtain (ambient) dimension-independent bounds that do not diverge with the number of iterations, recent works proposed to utilize the low-dimensional fractal structures generated by the SGD iterates whose complexity can be measured by a notion called the Hausdorff dimension [Fal14]. In this context, [ŞSDE20] showed that, under a continuous-time surrogate for SGD, the generalization error can be bounded by $\widetilde{O}(\sqrt{d_H/n})$, where $d_H$ denotes the Hausdorff dimension of the optimization trajectory. This result was later extended to discrete-time iterated function systems by [CDE+21]. Here, the Hausdorff dimension can be smaller than the ambient dimension [CDE+21], ultimately providing improved generalization bounds. However, both of these results are inherently asymptotic, and rely on opaque assumptions that are hard to verify in practice.

In this paper, we propose a new framework for deriving generalization bounds for the projected SGD with a constant step size and without requiring early stopping. Inspired by the works [ŞSDE20, CDE+21], our framework is based on a complexity measure of the trajectory of SGD, which can be quantified under standard verifiable conditions. Our contributions are as follows.

- **Localized $\varepsilon$-covers for SGD.** Our first principle contribution is a covering technique localized for the trajectories of SGD. This localization provides an algorithm-specific complexity measured by the covering number, which can have dimension-independent cardinality in contrast to standard covering arguments that result in exponential dimension dependency.
- **Generalization bounds for SGD.** Based on this localized covering, we establish dimension-independent generalization bounds for SGD, for a rich class of non-convex loss functions $f$ whose gradients can be approximated by that of a piecewise strongly convex and smooth function $h$, i.e. $\|\nabla f(\theta; z) - \nabla h(\theta; z)\| \leq \xi$ for some $\xi$. In particular, with high probability, we prove the bound

$$\left| \hat{F}(\theta^{(t)}) - F(\theta^{(t)}) \right| = O\left( \sqrt{\frac{\log n \log(nP)}{n}} + \xi \right), \tag{1.3}$$

  where $\theta^{(t)}$ denotes the parameter generated by $t$ SGD iterations for a sufficiently large $t$, and $P$ denotes the number of strongly convex pieces needed to approximate $f$. We further show that the gradient of any (piecewise) smooth function $f$ can be approximated with that of a piecewise strongly convex and smooth function, demonstrating the wide applicability of the bound (1.3). Finally in the special case $P = 1$ and $\xi = 0$, our result reduces to a non-asymptotic bound where the complexity is captured by the Hausdorff dimension of the invariant measure of SGD.
- **Improved bounds in specific models.** We employ the above result to derive generalization bounds in several statistical models trained by SGD, including multi-index linear models, multi-class support vector machines, and $K$-means clustering with both hard and soft label setups, improving the previously known state-of-the-art generalization error bounds in this context.

**Notation and problem setup.** For $k \in \mathbb{N}$, we denote $[k] := \{1, \ldots, k\}$ We use $\|\cdot\|$ to denote the $\ell_2$-norm. For $\varepsilon > 0$ and $\theta \in \mathbb{R}^d$, we use $\mathcal{B}_\varepsilon^d(\theta)$ to denote the $d$-dimensional closed $\ell_2$-ball of radius $\varepsilon$, centered at $\theta$. Given a set $\mathcal{S} \subseteq \mathbb{R}^d$ and $\varepsilon > 0$, we say $\mathcal{C}_\varepsilon \subseteq \mathbb{R}^d$ is an "$\varepsilon$-cover" of $\mathcal{S}$ if $\mathcal{S} \subseteq \bigcup_{\theta \in \mathcal{C}_\varepsilon} \mathcal{B}_\varepsilon^d(\theta)$.

Given a step size $\eta > 0$, an initial parameter $\theta^{(0)} \in \Theta$, and a randomly sampled index $i_t \in [n]$, the $t$-th iteration of the projected SGD performs the following update on the parameters

$$\theta^{(t)} = g_{i_t}(\theta^{(t-1)}) := \Pi_\Theta\big(\theta^{(t-1)} - \eta \nabla f(\theta^{(t-1)}; z_{i_t})\big) \quad \text{for} \ \ t = 1, 2, ..., \tag{1.4}$$

where $\Pi_\Theta(\theta) := \arg\min_{\theta' \in \Theta} \|\theta' - \theta\|$ denotes the Euclidean projection. The domain $\Theta$ is convex; thus, the projection operation is unique. We note that the projection is not needed under the presence of an $\ell_2$-regularizer and Lipschitz continuity; see Section 4 for more details. Throughout the paper, we use $\theta^{(t)} := g_{i_t} \circ \cdots \circ g_{i_1}(\theta^{(0)})$ for possibly random indices $i_1, \ldots, i_t$.

Lastly, we recall a few standard notions. $f : \Theta \to \mathbb{R}$ is called "$\alpha$-strongly convex" and "$\beta$-smooth" respectively if for all $\theta, \theta' \in \Theta$, the following conditions are satisfied

$$f(\theta) - f(\theta') - \nabla f(\theta')^\top (\theta - \theta') \geq \frac{\alpha}{2} \|\theta - \theta'\|^2, \quad \text{and} \quad \|\nabla f(\theta) - \nabla f(\theta')\| \leq \beta \|\theta - \theta'\|.$$

The function $f$ is called "convex" if it is 0-strongly convex.

## 2 Main results

We demonstrate our covering construction in Section 2.1 in the classical strongly convex and smooth case in which the localization argument can be simplified by the contractivity of SGD. We present our main generalization result on non-convex losses in Section 2.2, and its implications in Section 3.

### 2.1 A localized covering construction: Strongly convex and smooth case

To motivate our approach, let us first briefly discuss the limitations of prior methods that are based on uniform convergence of empirical processes. Given a set of parameters $\Theta \subseteq \mathcal{B}_R^d(0)$, let $\mathcal{C}_\varepsilon$ be an $\varepsilon$-cover of $\Theta$, i.e. $\Theta \subseteq \bigcup_{\phi \in \mathcal{C}_\varepsilon} \mathcal{B}_\varepsilon^d(\phi)$. Uniform convergence over $\Theta$ can be established with high probability by simply applying the union bound over $\mathcal{C}_\varepsilon$, which yields a generalization error bound depending on the cardinality of the cover $\sqrt{\log |\mathcal{C}_\varepsilon|}$. However, $\mathcal{C}_\varepsilon$ is typically independent of the algorithm being used, and standard $\varepsilon$-covers for $\Theta$ yield $|\mathcal{C}_\varepsilon| = (R/\varepsilon)^{\Omega(d)}$; thus, bounds based on covering numbers often grow with $\sqrt{d}$, which can be loose if $d$ is large. To overcome this issue, we use the contractive properties of SGD in the strongly convex and smooth case and *localize* the $\varepsilon$-cover. Namely, instead of covering the entire feasible set $\Theta$, we construct a cover that contains only the points that can be reached by SGD trajectories, resulting in a covering number that is independent of the ambient dimension $d$. We introduce the following sets produced by SGD trajectories.

**Definition 1.** *We define the following two subsets of $\Theta$.*
- *The set of points that can be reached by $T$ SGD iterations initialized at $\theta^{(0)} \in \Theta$,*
$$\Psi_T(\theta^{(0)}) := \{g_{i_T} \circ \cdots \circ g_{i_1}(\theta^{(0)}) : i_1, \ldots, i_T \in [n]\}.$$
- *The set of points that can be reached by any $t \geq T$ SGD iterations initialized at $\theta^{(0)} \in \Theta$,*
$$\Psi_{\geq T}(\theta^{(0)}) := \bigcup_{t \geq T} \Psi_t(\theta^{(0)}).$$

For $\gamma \in (0, 1)$, a function $g : \Theta \to \Theta$ is called "$\gamma$-contractive" if for all $\theta, \theta' \in \Theta$, it satisfies $\|g(\theta) - g(\theta')\| \leq \gamma \|\theta - \theta'\|$. In the constant step-size case, SGD iterates converge to a distribution instead of a single point [DDB20], but their contractivity can still provide the following localization: all possible SGD iterates after sufficiently many iterations can be $\varepsilon$-covered by $n^{O(\log(1/\varepsilon))}$ points.

**Lemma 2.1.** *Suppose that $g_1, g_2, \ldots$ are $\gamma$-contractive for some $\gamma \in (0, 1)$. Then, for any initialization $\theta^{(0)} \in \Theta \subseteq \mathcal{B}_R^d(0)$ and for any $\varepsilon > 0$, for $T := T_\varepsilon = \max\left\{\left\lceil \frac{\log(R/\varepsilon)}{\log(1/\gamma)} \right\rceil, 0\right\}$, we have*
$$\Psi_{\geq T}(\theta^{(0)}) \subseteq \bigcup_{\phi \in \Psi_T(0)} \mathcal{B}_\varepsilon^d(\phi).$$

*Proof.* For $t \geq T$, let $\theta^{(t)} \in \Psi_{\geq T}(\theta^{(0)})$ such that $\theta^{(t)} := g_{i_t} \circ \cdots \circ g_{i_1}(\theta^{(0)})$ for some $i_1, \ldots, i_t \in [n]$. Let $\phi := g_{i_t} \circ \cdots \circ g_{i_{t-T+1}}(0)$ and notice that $\phi \in \Psi_T(0)$ by construction. Then for $T := T_\varepsilon$ and $\theta^{(t-T)} := g_{i_{t-T}} \circ \cdots \circ g_{i_1}$, we have
$$\|\theta^{(t)} - \phi\| \leq \gamma^T \|\theta^{(t-T)} - 0\| \leq \gamma^T R \leq \varepsilon,$$
since each $g_i$ is $\gamma$-contractive and $\|\theta^{(t-T)}\| \leq R$ by (1.4). This implies $\theta^{(t)} \in \bigcup_{\phi \in \Psi_T(0)} \mathcal{B}_\varepsilon^d(\phi)$. $\square$

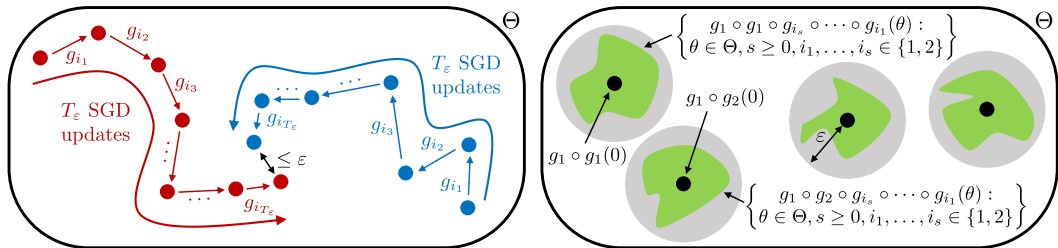

Figure 1: Left: illustration of $T = T_\varepsilon$ coupled projected SGD updates from two distinct points. Right: illustration of our localized cover $\Psi_T$ covering $\Psi_{\geq T}$ with $T = 2$ and $n = 2$ and $|\Psi_T| = n^T = 4$.

For strongly convex and smooth $f(\cdot; z)$, an SGD update $g_i$ with a sufficiently small step size is contractive [DDB20]; that is, applying $g_i$ to any two points decreases the distance between them (see Appendix A.1 for a formal derivation). Therefore, for any $\varepsilon > 0$, there exists $T := T_\varepsilon$ such that applying $T$ *synchronously coupled* SGD updates that use the same sample at each iteration can make the distance between the initial points smaller than $\varepsilon$; see Figure 1 (left). This observation implies that the set $\Psi_T(0)$ of all parameters that can be generated by $T$ SGD updates when initialized at the origin $\varepsilon$-*covers* the set $\Psi_{\geq T}(\theta^{(0)})$ of all parameters that can be generated by any $t \geq T$ SGD updates for an arbitrary initialization $\theta^{(0)}$; see Figure 1 (right). In contrast to algorithm-independent covers of $\Theta$ that scale with $|\mathcal{C}_\varepsilon| = (R/\varepsilon)^{\Omega(d)} = e^{\widetilde{\Omega}(d)}$, we obtain $|\Phi_T(0)| \leq n^T = e^{\widetilde{O}(1)}$ which is independent of the dimension $d$ and only polynomial in the number of samples $n$.

We make the following additional assumptions on the loss function $f$.

**Assumption 1** (Weak Lipschitz continuity). *For $L > 0$, there exists $h : \Theta \to \mathbb{R}$ such that for all $\theta, \theta' \in \Theta$ and $z \in \mathcal{Z}$, $|f(\theta; z) - h(\theta) - (f(\theta'; z) - h(\theta'))| \leq L\|\theta - \theta'\|$.*

Notice that $h = 0$ reduces to the classical Lipshitz continuity, but the above condition is more general.

**Assumption 2** (Bounded deviation). *For $B > 0$, for all $\theta \in \Theta$, $\left| \sup_{z \in \mathcal{Z}} f(\theta; z) - \inf_{z \in \mathcal{Z}} f(\theta; z) \right| \leq B$.*

We note that Assumptions 1 & 2 (or their variants) both appear in several algorithmic stability-based results; see e.g. [HRS15, Thm 3.10]. We also highlight that these conditions are invariant to adding regularizers (e.g. consider $f(\theta; z) \leftarrow f(\theta; z) + \frac{\lambda}{2}\|\theta\|^2$) which will be useful in Section 3.

**Theorem 2.1.** *Suppose that Assumptions 1 & 2 hold and there exist $\alpha, \beta > 0$ such that $f(\cdot; z)$ is $\alpha$-strongly convex and $\beta$-smooth on $\Theta \subseteq \mathcal{B}_R^d(0)$ for all $z \in \mathcal{Z}$. For any $\eta \in (0, 2/\beta)$, let $\gamma := \sqrt{1 - 2\alpha\eta + \alpha\beta\eta^2}$, $T := \max\left\{ \left\lceil \frac{\log(2LRn)}{\log(1/\gamma)} \right\rceil, 0 \right\}$. Then, with probability at least $1 - \delta$, for any $\theta^{(0)} \in \Theta$, $t \geq T$, and $i_1, \ldots, i_t \in [n]$, SGD iterate $\theta^{(t)}$ satisfies*

$$\left| \hat{F}(\theta^{(t)}) - F(\theta^{(t)}) \right| \leq \frac{BT + 1}{n} + B\sqrt{\frac{T \log n + \log(2/\delta)}{2n}}. \tag{2.1}$$

**Remark.** A few remarks are in order. For stochastic convex optimization, the best achievable bound via uniform convergence over $\Theta$ is worse than the algorithmic stability-based bounds for SGD by a factor of $\sqrt{d}$ [SSSSS09, Fel16, FV19]; however, by localizing the uniform convergence argument, we are able to obtain a dimension-independent bound. We note that this high-probability result is still not directly comparable to most stablity-based bounds which are given in expectation. An exception is [FV19, Thm 4.5] which obtains near-optimal bounds for SGD in the convex case by tuning the number of iterations to be taken, obtaining a rate of $O(\log(n/\delta)^2 \log(n)/\sqrt{n})$. In contrast, our bound reads $O(\sqrt{\log(n)^2 \log(1/\delta)/n})$ which is better by a logarithmic factor and holds for any sufficiently large number of SGD iterations. We emphasize that the bound (2.1) holds for *the union of all trajectories of SGD* generated by $t \geq T$ iterations, whereas the stability-based results often consider a single parameter generated by SGD. A similar setup is considered in Corollary 2.1 where we establish improved bounds on the generalization error removing the logarithmic factor. Lastly, we note that our localized covering can also be used for deriving bounds in expectation; see Appendix D.

The proof of Theorem 2.1 follows from three steps. $i)$ Lemma 2.1 implies that any $\theta^{(t)}$ for $t \geq T$ can be approximated by some parameter $\theta^{(T)}$ generated by $T$ SGD updates initialized at 0. $ii)$ $\hat{F}(\theta^{(T)})$

concentrates around $F(\theta^{(T)})$ since $\theta^{(T)}$ is *almost* independent of the samples, i.e. it depends on at most $T = O(\log(n))$ of $n$ samples. *iii)* The empirical process $|\hat{F}(\theta^{(T)}) - F(\theta^{(T)})|$ uniformly converges over the set $\Psi_T(0)$ which has a dimension-free cardinality. See Appendix B.2 for details.

While the bound in Theorem 2.1 holds uniformly over all possible initializations, it can be tightened by considering a single realization of $\theta^{(0)}$ in the following corollary.

**Corollary 2.1.** *Assume the setup in Theorem 2.1. Then, for any $\theta^{(0)} \in \Theta$, $t \geq T$, and $i_1, \ldots, i_t \in [n]$, with probability at least $1 - \delta$, we have*

$$\left| \hat{F}(\theta^{(t)}) - F(\theta^{(t)}) \right| \leq \frac{BT + 1}{n} + B\sqrt{\frac{\log(2/\delta)}{2n}}.$$

For sufficiently large $n$, the bound in Corollary 2.1 coincides with the tight concentration bound at a fixed $\theta \in \Theta$; however, it has the limitation that at least $t \geq T$ SGD updates are required. This is remedied in the next result which does not require strong convexity and smoothness.

**Corollary 2.2.** *Suppose that Assumption 2 holds. Then, for any $\theta^{(0)} \in \Theta$, $t \geq 0$, and $i_1, \ldots, i_t \in [n]$, with probability at least $1 - \delta$, we have*

$$\left| \hat{F}(\theta^{(t)}) - F(\theta^{(t)}) \right| \leq \frac{Bt}{n} + B\sqrt{\frac{\log(2/\delta)}{2n}}.$$

The bound in Corollary 2.2 requires early stopping since it diverges as $t \to \infty$. However, since Corollary 2.2 only requires Assumption 2, it can be easily combined with other results, e.g. Corollary 2.1, to provide a generalization bound that holds for any number of SGD iterations. The proofs of Corollaries 2.1 and 2.2 are presented in Appendices B.3 and B.4, respectively.

**Relation to fractal dimension.** There is an interesting connection to be made between Theorem 2.1 and the Hausdorff dimension of the support $\mu$ of the stationary distribution of (projected) SGD. For example, suppose that $\eta \in (0, 1)$, $\Theta = \mathcal{B}_R^d(0)$, $f(\theta; z) = \frac{1}{2}\|\theta - \theta_z^*\|^2$ for some $\theta_z^* \in \mathcal{B}_R^d(0)$, and $\|\theta_{z_i}^* - \theta_{z_j}^*\| \geq 2\gamma R$ for all $i \neq j$. Here, the last assumption can be satisfied with high probability if $d, \eta$ are large enough and $\theta_z^* \sim N(0, \sigma^2 I)$, i.e. zero-mean Gaussian with covariance $\sigma^2 I$; see e.g. [PLYS21, Appendix C]. Under these assumptions, the bound in Theorem 2.1 can be reformulated as

$$\left| \hat{F}(\theta^{(t)}) - F(\theta^{(t)}) \right| \leq \frac{BT + 1}{n} + B\sqrt{\frac{\left\lceil d_H + \frac{\log 2LR}{\log(1/\gamma)} \right\rceil \log n + \log(2/\delta)}{2n}}, \tag{2.2}$$

where $d_H$ denotes the Hausdorff dimension of $\mu$. The precise definition of $d_H$ and the derivation of (2.2) are presented in Appendix B.5. In (2.2), $d_H$ replaces the ambient dimension $d$ that appear in algorithm-independent bounds [SSBD14]. We note that such a connection between the fractal dimension and generalization bounds has been also studied in [ŞSDE20, CDE$^+$21]; however, their bounds are asymptotic and require non-trivial assumptions that are not easy to verify in practice.

**Generalization due to contractivity.** Contractivity of SGD has also been used to derive generalization bounds in the concurrent work [KWS22]. Although the localized covering construction in Lemma 2.1 relies on the same principle, our results differ from those in [KWS22] on two key aspects. First and foremost, as we shall see in Section 2.2, the localized covering argument used in Lemma 2.1 can also be applied to the non-convex case in which the uniform stability-based argument used in [KWS22] provably breaks down (Appendix F); thus, extending the results of [KWS22] to cover non-convex objectives is highly non-trivial. Further, the generalization bounds in [KWS22] are provided in expectation, i.e. $|\mathbb{E}[\hat{F}(\theta^{(t)}) - F(\theta^{(t)})]| \leq O(1/n)$, whereas we provide high-probability bounds, i.e. $|\hat{F}(\theta^{(t)}) - F(\theta^{(t)})| \leq \widetilde{O}(1/\sqrt{n})$. When translated to bounds in expectation, our results read $|\mathbb{E}[\hat{F}(\theta^{(t)}) - F(\theta^{(t)})]| \leq \widetilde{O}(1/n)$ and $\mathbb{E}[|\hat{F}(\theta^{(t)}) - F(\theta^{(t)})|] \leq \widetilde{O}(1/\sqrt{n})$; see Appendices D & E.

## 2.2 Non-convex case: Perturbations of piecewise strongly convex and smooth functions

Algorithmic stability technique yields (near) optimal rates for strongly convex objectives; however, when applied to non-convex functions, the resulting bounds often diverge with the number of SGD iterations [HRS15, LLQ20]. The localized covering construction introduced in the previous section remedies this issue, providing more stable generalization bounds. Specifically, we establish a

dimension-independent generalization bound for functions that are finite perturbations of piecewise strongly convex and smooth functions. We further prove an approximation result and show that any smooth non-convex function can be approximated with a piecewise strongly convex function. Since piecewise strongly convex and smooth functions may not be differentiable on the entire $\Theta$, we will use the auxiliary gradient in Definition 2 as a surrogate for the SGD update (1.4).

**Definition 2.** *$f$ is "piecewise $\alpha$-strongly convex and $\beta$-smooth with $P$ pieces on $\Theta$" if there exists a partition $\{\mathcal{P}_1, \ldots, \mathcal{P}_P\}$ of $\Theta$ and $\alpha$-strongly convex and $\beta$-smooth functions $h_1, \ldots, h_P$ on $\Theta$ such that $f = h_p$ on $\mathcal{P}_p$ for all $p \in [P]$. We also define $\nabla f(\theta) := \nabla h_p(\theta)$ if $\theta \in \mathcal{P}_p$.*

Piecewise strongly convex and smooth objectives are widely used in machine learning applications. For example, the objective of learning a single layer of a ReLU network with $\ell_2$-regularization is piecewise strongly convex and smooth. Furthermore, the objective of learning an entire ReLU network is also piecewise strongly convex and smooth on small loss regions [Mil19]. Such observations easily extend to more general settings, e.g. an objective function defined as the summation of a piecewise linear loss (e.g. hinge loss) and $\ell_2$-regularization. However, if we further allow for finite perturbations, any piecewise smooth non-convex function can be covered within this framework.

**Proposition 2.1.** *For any piecewise $\beta'$-smooth $f$ with $Q$ pieces on $\Theta \subset \mathcal{B}_R^d(0)$, and for any $\xi > 0$ and $0 < \alpha \leq \beta$, there exists a piecewise $\alpha$-strongly convex and $\beta$-smooth $h$ with at most $Q(3(\beta + \beta')R/\xi)^d$ pieces such that $\|\nabla f(\theta) - \nabla h(\theta)\| \leq \xi$ for all $\theta \in \Theta$.*

For this general class of non-convex functions, we derive the following generalization bound.

**Theorem 2.2.** *Suppose that Assumptions 1 & 2 hold and $\Theta \subseteq \mathcal{B}_R^d(0)$. Suppose further that there exists $h : \Theta \times \mathcal{Z} \to \mathbb{R}$ such that for any $z \in \mathcal{Z}$, $h(\,\cdot\,; z)$ is piecewise $\alpha$-strongly convex and $\beta$-smooth with $P$ pieces on $\Theta$ satisfying*

$$\|\nabla f(\theta; z) - \nabla h(\theta; z)\| \leq \xi, \tag{2.3}$$

*for all $\theta \in \Theta$. For any $\eta \in (0, 2/\beta)$, let $\gamma := \sqrt{1 - 2\alpha\eta + \alpha\beta\eta^2}$. Then given $T \in \mathbb{N}$, with probability at least $1 - \delta$, for any $\theta^{(0)} \in \Theta$, $t \geq T$, and $i_1, \ldots, i_t \in [n]$, we have*

$$\left| \hat{F}(\theta^{(t)}) - F(\theta^{(t)}) \right| \leq \frac{BT}{n} + B\sqrt{\frac{T \log(nP) + \log(2/\delta)}{2n}} + 2L\left(\gamma^T R + \frac{1 - \gamma^T}{1 - \gamma}\eta\xi\right)$$

**Remark.** The bound above is stated in full generality and holds for any SGD iterate $t \geq T$ and any choice of $T \geq 1$. However, to obtain a meaningful generalization bound, one may choose, for example $T = O(\log(nR)/\log(\gamma^{-1}))$. In the case that $\gamma, B, L = \Theta(1)$, the bound simplifies to

$$\left| \hat{F}(\theta^{(t)}) - F(\theta^{(t)}) \right| \leq O\left(\sqrt{\frac{\log(nR) \log(nP) + \log(1/\delta)}{n}} + \xi\right). \tag{2.4}$$

Here, the first term in the bound is logarithmic in the number of pieces $P$ and the last term scales linearly with the approximation error $\xi$. Thus, the generalization bound depends on the trade-off between the complexity of $h$ through $P$, and how well $\nabla h$ approximates $\nabla f$ through $\xi$. In this regime, in contrast to algorithmic stability-based bounds, the above result does not grow with the number of SGD iterations, i.e. early stopping is not required for generalization in Theorem 2.2. We finally note that Theorem 2.2 holds for any $f$ with an *auxiliary gradient* $\nabla f$ satisfying (2.3).

In light of Proposition 2.1, any piecewise smooth function can be approximated by a piecewise strongly convex and smooth function; in the worse case, the number of pieces $P$ is at most $e^{\widetilde{\Omega}(d)}$. Therefore in this pessimistic case, Theorem 2.2 recovers the classical algorithm-independent covering bound $\widetilde{O}(B\sqrt{d/n})$ by choosing $\xi = O(1/(L\sqrt{n}))$ and $T = \Theta(\log(LRn))$. However, any value of $P$ that is sub-exponential in dimension yields improved generalization bounds. In particular in the next section, we consider certain (non-convex) statistical models and carefully design $h$ so that Theorem 2.2 improves the existing generalization error bounds. In contrast, uniform stability-based bounds for piecewise strongly convex and smooth functions are in general $\Omega(1)$ after sufficiently many SGD iterations; see Appendix F. That is, the contractivity-based bounds in [KWS22] cannot be directly extended to piecewise contractivity (2.3).

The proof of Theorem 2.2 relies on a modified version of the covering construction presented in Section 2.1. First, we define the auxiliary parameter update $g_{i,p}(\theta) := \theta - \nabla h_p(\theta; z_i)$ where $h_p(\,\cdot\,; z_i)$

denotes the strongly convex and smooth function satisfying $\nabla f(\cdot\,; z_i) \approx \nabla h_p(\cdot\,; z_i)$ on the $p$-th piece. We show that $\Psi'_T(0) := \{g_{i_T, p_T} \circ \cdots \circ g_{i_1, p_1}(0) : i_1, \ldots, i_T \in [n], p_1 \ldots, p_T \in [P]\}$ $\varepsilon$-covers $\Psi_{\geq T}(\theta^{(0)})$ for any $\theta^{(0)} \in \Theta$ as in Lemma 2.1. Since $|\Psi'_T| \leq (nP)^T$, applying a concentration inequality and the union bound over the localized cover yields Theorem 2.2. Formal proofs of Proposition 2.1 and Theorem 2.2 are provided in Appendices B.6 and B.7, respectively.

## 3 Applications

In this section, we use our localized covering to prove generalization bounds for multi-index linear models, multi-class support vector machines, and $K$-means clustering for both hard and soft label setups, trained by SGD, improving the state-of-the-art results known for these models.

### 3.1 Multi-index linear models

Given a sample $z = (y, x) \in \mathcal{Y} \times \mathcal{B}^d_{R_x}(0)$, consider the $\ell_2$-regularized loss in a *multi-index model* with $K$ indices parameterized by $\theta = (\theta_j)_{j=1}^K \in \mathbb{R}^{d \times K}$

$$f(\theta; z) := \ell(\theta_1^\top x, \ldots, \theta_K^\top x; y) + \sum_{j=1}^K \frac{\lambda}{2} \|\theta_j\|^2. \tag{3.1}$$

Characterizing the generalization properties of multi-index models is an important problem with many applications including regression, classification, dimension reduction, and learning a single-layer of a neural network. In the following theorem, we derive a generalization bound for multi-index models by approximating each $f(\cdot\,; z)$ with a piecewise strongly convex and smooth function, and then applying Theorem 2.2. The proof is given in Appendix C.1.

**Theorem 3.1.** *Let $\theta = (\theta_j)_{j=1}^K$ and $\Theta = \prod_{j=1}^K \Theta_j$ for some convex $\Theta_j \subseteq \mathcal{B}^d_R(0)$. Suppose that $f$ satisfies (3.1), Assumptions 1 & 2 hold, and $\ell(\cdot, \ldots, \cdot\,; y)$ is piecewise $\beta$-smooth with $Q$ pieces on $\Theta$ for all $y \in \mathcal{Y}$. For any $\eta \in (0, 2/\lambda)$, let $\gamma := |1 - \eta\lambda|$, $T := \max\left\{\left\lceil \frac{\log(3LRn)}{\log(1/\gamma)} \right\rceil, 0\right\}$, and $P := \mathtt{poly}(\beta, \eta, K, L, R, R_x, T, n)$. Then, with probability at least $1 - \delta$, for any $\theta^{(0)} \in \Theta$, $t \geq T$, and $i_1, \ldots, i_t \in [n]$, we have*

$$\left|\hat{F}(\theta^{(t)}) - F(\theta^{(t)})\right| \leq \frac{BT + 1}{n} + B\sqrt{\frac{T \log(nP^K Q) + \log(2/\delta)}{2n}}.$$

Generalization behavior of multi-index models has received considerable attention; a subset of notable results include [Gue02, Zha04, JKZ+12, CMR13, LDZK19]. Under the same conditions of Theorem 3.1, existing state-of-the-art bounds scale at least linearly in $\sqrt{d}$ or $K$, while our result is dimension-free and scales with $\sqrt{K}$. Specifically, the result in [LDZK19] translated to our setting reads ($\ell$ is $L$-Lipschitz, and $f(\theta; z) \in [0, B]$ which are stronger conditions than Assumptions 1 & 2)

$$\left|\hat{F}(\theta) - F(\theta)\right| \leq \widetilde{O}\left(\frac{KLRR_x + B}{\sqrt{n}}\right), \tag{3.2}$$

for all $\theta \in \Theta$, which is obtained via [LDZK19, Cor 3 & 9] with $\Lambda \leftarrow R$ and $p \leftarrow \infty$. While the bound in (3.2) is linear in $KLRR_x$, our bound is linear in $\sqrt{K}$ and logarithmic in $L, R, R_x$. The significance of this improvement can be better seen, for example, when learning the first layer of neural networks, for which $L$ can be very large. Moreover, if $x \sim N(0, I)$, then $\|x\| = \widetilde{\Omega}(\sqrt{d})$ with high probability; that is, (3.2) is linear in $R_x = \Omega(\sqrt{d})$ but ours only scales with $\sqrt{\log d}$, which can make a significant difference especially in the overparameterized regime. We should note that compared to our bound, (3.2) does not require the smoothness of $\ell$.

For a specific application of Theorem 3.1, consider the multi-class support vector machines, i.e. for $\mathcal{Y} = [K]$ and $\rho : [-RR_x, RR_x] \to [0, B]$ is $L$-Lipschitz and $\beta$-smooth, the objective is given as

$$f(\theta; z) = \max_{y' \neq y} \rho\left(\theta_y^\top x - \theta_{y'}^\top x\right) + \frac{\lambda}{2} \sum_{j=1}^K \|\theta_j\|^2.$$

Theorem 3.1 provides a generalization bound of $\widetilde{O}(\sqrt{K/n})$ that improves the existing bounds which scale at least linearly with $K$ and/or $\sqrt{d}$ [Zha04, DSBDSS15, LDBK15, LDZK19]; see e.g. [LDZK19, Sec 2.1] for similar related results.

## 3.2 $K$-means clustering

We can also provide generalization error bounds for the problem of $K$-means clustering in both hard and soft label setups. As before, we consider SGD as an optimizer which has been the focus of many works in this context [BB94, Scu10, TM17]. We assume throughout this section that the samples are supported on a bounded domain, and the SGD iterations are applied without projection.

Generalization properties of $K$-means clustering has been studied for decades [Ant05, Lev13, TTJC15, TM16]. While most existing bounds are at least linear in $K$, [LL21] recently provided an improved bound in the hard label setup, which is of order $(K \log^3 n/n)^{1/2}$ for bounded inputs. However, state-of-the-art bounds in the soft label setup were still linear in $K$. Below, we establish a bound of order $\sqrt{K \log n/n}$ in the soft label setup, which is the first bound that is sublinear in $K$. We further improve the bounds of [LL21] in the hard label setup, but by a logarithmic factor.

**Soft $K$-means clustering.** While the hard $K$-means clustering assigns exactly one cluster to each point using one-hot encoding, the soft $K$-means clustering allows *soft labels* within the probability simplex. Specifically, given $\zeta > 0$ and the samples $z_1, \ldots, z_n$, the soft $K$-means clustering algorithm alternates between updating the soft labels $(w_j(z_i; \theta))_{j=1}^{K}$ of $z_i$ for all $i \in [n]$, and estimating the cluster centers $\theta = (\theta_j)_{j=1}^{K} \in (\mathcal{B}_R^d(0))^K$ using the classical update rule [Mac03]

$$w_j(z_i; \theta) \leftarrow \frac{\exp(-\zeta\|\theta_j - z_i\|^2)}{\sum_{k=1}^{K} \exp(-\zeta\|\theta_k - z_i\|^2)}, \qquad \theta_j \leftarrow \frac{\sum_{i=1}^{n} w_j(z_i; \theta) z_i}{\sum_{i=1}^{n} w_j(z_i; \theta)}. \tag{3.3}$$

Here, this procedure is equivalent to a special case of expectation-maximization algorithm, which converges to a local minimum of the following objective (see Appendix C.2 for details)

$$\hat{F}(\theta) = \frac{1}{n} \sum_{i=1}^{n} \left\{ f(\theta; z_i) := -\frac{1}{\zeta} \log \left( \sum_{j=1}^{K} \exp\left( -\zeta\|\theta_j - z_i\|^2 \right) \right) \right\}. \tag{3.4}$$

Instead of running the standard alternating procedure (3.3), we directly minimize (3.4) using SGD and derive generalization bounds for the soft $K$-means clustering by approximating the objective (3.4) with piecewise strongly convex and smooth functions. The proof is presented in Appendix C.3.

**Theorem 3.2.** *For $B := 4(R+1)^2$ and for any $\eta \in (0, Ke^{-\zeta B})$, let $\gamma := \sqrt{1 - \frac{4\eta e^{-\zeta B}}{K} + \frac{4\eta^2}{K^2}}$, $L := \frac{4R}{\sqrt{K}} e^{\zeta B}$, $T := \max\left\{ \left\lceil \frac{\log(3LRn)}{\log(1/\gamma)} \right\rceil, 0 \right\}$, and $P := \mathtt{poly}(\eta, \zeta, B, K, R, T, n, e^{\zeta B})$. Then, with probability at least $1 - \delta$, for any $\theta^{(0)} \in \Theta$, $t \geq T$, and $i_1, \ldots, i_t \in [n]$, we have*

$$\left| \hat{F}(\theta^{(t)}) - F(\theta^{(t)}) \right| \leq \frac{BT+1}{n} + B\sqrt{\frac{T \log(nP^K) + \log(2/\delta)}{2n}}.$$

The above generalization bound scales with $\sqrt{K}$, which improves the previously known bound $\widetilde{O}(K/\sqrt{n})$ for clustering with soft labels [LL21]. Theorem 3.2 also implies $\widetilde{O}(\sqrt{K/n})$ bound under $R, \zeta = \Theta(1)$ and $\eta = \Theta(K)$, which coincides with the best known rate in the hard label setup. Here, the choice of $\eta = \Theta(K)$ may look odd; however, it is indeed practical since the derivative of the objective function $f$ scales with $1/K$, i.e. $\|\partial f(\theta; z)/\partial \theta_j\| \leq 4Re^{\zeta B}/K$.

**Hard $K$-means clustering.** Under the same setup as before, we minimize the objective function

$$\hat{F}(\theta) = \frac{1}{n} \sum_{i=1}^{n} \left\{ f(\theta; z_i) := \min_{j \in [K]} \|\theta_j - z_i\|^2 \right\}. \tag{3.5}$$

Note that (3.5) coincides with the soft $K$-means objective (3.4) as $\zeta \to \infty$ if $\|\theta_j - z_i\| \neq \|\theta_k - z_i\|$ for all $i, j, k$. Since $f(\theta; z) = \min_{j \in [K]} \|\theta_j - z\|^2$ may not be differentiable, we use an auxiliary gradient at a non-differentiable $\theta$, i.e. for a randomly or deterministically chosen $\mathcal{S} \subseteq \arg\min_{j \in [K]} \|\theta_j - z\|^2$,

$$\frac{\partial}{\partial \theta_j} f(\theta; z) := \begin{cases} 0 & \text{if } j \notin \mathcal{S} \\ 2(\theta_j - z) & \text{if } j \in \mathcal{S} \end{cases}.$$

For example, one may choose a single cluster index ($|\mathcal{S}| = 1$) and compute the corresponding gradient. The following result characterizes the generalization of hard $K$-means clustering trained by SGD using this auxiliary gradient. Its proof is deferred to Appendix C.4.

**Theorem 3.3.** *For any $\eta \in (0,1)$, let $B := 4R^2$, $\gamma := |1 - 2\eta|$, $T := \max\left\{ \left\lceil \frac{\log(16\sqrt{K}R^2 n)}{\log(1/\gamma)} \right\rceil, 0 \right\}$. Then, given $\theta^{(0)} \in \Theta$, the following bound holds with probability at least $1 - \delta$, for any $t \geq 0$ and $i_1, \ldots, i_t \in [n]$, we have*

$$\left| \hat{F}(\theta^{(t)}) - F(\theta^{(t)}) \right| \leq \frac{BKT + 1}{n} + B\sqrt{\frac{KT \log(2n) + \log(2/\delta)}{2n}}.$$

We note that under $\eta, R = \Theta(1)$ and $K = O(n)$, Theorem 3.3 provides $O((K \log^2 n/n)^{1/2})$ generalization bound which improves the bound given by [LL21], but this time by a logarithmic factor.

## 4   Discussions

**Beyond (piecewise) strong convexity and smoothness.** The localized covering construction we utilized is essentially based on the (piecewise) contractivity of SGD updates for (piecewise) strongly convex and smooth functions. However, local strong convexity is by no means necessary and this analysis can be extended to a broader class of objective functions.

Consider, for example, an objective function $f$ that is uniformly convex [DN21] and has Hölder continuous gradient [Nes15]. Then, for any $\varepsilon > 0$, there exists a step size $\eta > 0$ and $T = T_\varepsilon$ such that applying $T$ synchronously coupled SGD updates on any two points makes the distance between them smaller than $\varepsilon$. Consequently, an analog of Lemma 2.1 can be established under this setup as well, for a properly chosen $\eta, \varepsilon, T$. We highlight that this class of functions is already covered in our framework via Proposition 2.1; however, an analysis based on the actual curvature (as opposed to a piecewise approximation) of the objective may provide tighter generalization bounds.

**Extension to SGD without projection.** Projection operation is only needed to ensure that the SGD iterates stay bounded. Indeed, the projection is not needed in the presence of explicit regularization, or under a dissipativity-type condition on the objective [RRT17, EMS18, YBVE21, EH21]. For example, if $f(\cdot\,; z)$ is Lipschitz continuous, it is a straightforward exercise to show that SGD iterations are bounded in the presence of $\ell_2$-regularization and bounded initialization.

**Extension to mini-batch setup and different sampling schemes.** The results we presented can be easily extended to the mini-batch setting. To see this, note that the localization is based on the (piecewise) contractivity of a single iteration of the algorithm, and we have that if all $g_1, \ldots, g_n$ are contractive, then an average of their subset is also contractive. Moreover, our generalization bounds hold for any sampling scheme, e.g. sampling without replacement, random shuffling, data-dependent sampling, since the covering construction is based on an $\varepsilon$-cover of *the union of all possible trajectories of SGD*, which is independent of the underlying sampling scheme.

**Extension to contractive stochastic optimization methods.** In this paper, our main focus was the SGD algorithm and its generalization properties. However, it is straightforward to adapt our framework to a general stochastic optimization setup. For example, the next result follows from the identical steps leading to Theorem 2.2; hence, its proof is omitted.

**Definition 3.** *An "iterative stochastic algorithm" using $g : \Theta \times \mathcal{Z} \to \Theta$ performs the update $\theta^{(t)} = g_{i_t}(\theta^{(t-1)}) := g(\theta^{(t-1)}; z_{i_t})$ at iteration $t$, for a random sample $i_t \in [n]$.*

**Definition 4.** *$g : \Theta \to \Theta$ is "piecewise $\gamma$-contractive with $P$ pieces on $\Theta$" if there exists a partition $\mathcal{P}_1, \ldots, \mathcal{P}_P$ of $\Theta$ and $\gamma$-contractive $h_1, \ldots, h_P$ on $\Theta$ such that $g = h_p$ on $\mathcal{P}_p$ for all $p \in [P]$.*

**Theorem 4.1.** *Suppose that Assumptions 1 & 2 hold, $\Theta \subset \mathcal{B}_R^d(0)$, and an iterative stochastic algorithm $g$ is given. Suppose further that there exists $h : \Theta \times \mathcal{Z} \to \Theta$ such that for any $z \in \mathcal{Z}$, $h(\cdot\,; z)$ is piecewise $\gamma$-contractive with $P$ pieces on $\Theta$ satisfying*

$$\|g(\theta; z) - h(\theta; z)\| \leq \xi$$

*for all $\theta \in \Theta$. Choose $T \in \mathbb{N}$. Then with probability at least $1 - \delta$, for any $\theta^{(0)} \in \Theta$, $t \geq T$, and $i_1, \ldots, i_t \in [n]$, we have*

$$\left| \hat{F}(\theta^{(t)}) - F(\theta^{(t)}) \right| \leq \frac{BT}{n} + B\sqrt{\frac{T \log(nP) + \log(2/\delta)}{2n}} + 2L\left( \gamma^T R + \frac{1 - \gamma^T}{1 - \gamma} \xi \right).$$

We note that a concurrent work by [KWS22] requires *global* contractivity and their bounds are in expectation, whereas Theorem 4.1 is a high-probability statement which only requires *piecewise* contractivity as stated in Definition 4.

**Limitation of our results.** We outline a few limitations of our current analysis. In this paper, we only considered the generalization error (also referred to as the generalization gap), and our main result in the non-convex regime cannot be easily translated to a bound on the excess risk. This additional step would require a bound on the optimization error; nevertheless, establishing such bounds in the non-convex regime is highly non-trivial. It is worth highlighting that without assuming convexity, algorithmic stability-based bounds also suffer from this limitation [HRS15, FV19, BKZ20].

Because of the shared contractivity parameter $\gamma$ across different iterations and pieces, our method cannot be easily extended to varying step size. This is indeed a limitation of our analysis as varying step size schedules are oftentimes used in practice. We leave this extension as a future work.

## Acknowledgments

SP was supported by Institute of Information & communications Technology Planning & Evaluation (IITP) grant funded by the Korea government (MSIT) (No. 2019-0-00079, Artificial Intelligence Graduate School Program, Korea University) and Basic Science Research Program through the National Research Foundation of Korea (NRF) funded by the Ministry of Education (2022R1F1A1076180). US was funded in part by the French government under management of Agence Nationale de la Recherche as part of the "Investissements d'avenir" program, reference ANR-19-P3IA-0001 (PRAIRIE 3IA Institute), and the European Research Council Starting Grant DYNASTY – 101039676. MAE was supported by NSERC Grant [2019-06167], CIFAR AI Chairs program, and CIFAR AI Catalyst grant.

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
