# A   Technical results

All results presented in this section are standard. However, we provide their proofs for convenience.

**Lemma A.1.** *Let $\mathcal{S}$ be a Hibert space associated with the norm $\|\cdot\|$ induced by the inner product $\langle\cdot,\cdot\rangle$. Let $\mathcal{C} \subset \mathcal{S}$ be a convex set and $\Pi_{\mathcal{C}}(x) := \arg\min_{z\in\mathcal{C}} \|x - z\|$ be a projection of $x$ onto $\mathcal{C}$. Then,*

$$\|\Pi_{\mathcal{C}}(x) - \Pi_{\mathcal{C}}(y)\| \le \|x - y\| \quad \forall x, y \in \mathcal{S}.$$

*Proof.* Since $\mathcal{C}$ is convex, $\Pi_{\mathcal{C}}(x)$ is well-defined. From the definition of the projection and the convexity of $\mathcal{C}$, we have

$$\begin{aligned}
\|\Pi_{\mathcal{C}}(x) - x\|^2 &\le \left\|\big((1-t)\Pi_{\mathcal{C}}(x) + t\Pi_{\mathcal{C}}(y)\big) - x\right\|^2 \\
&= \left\|\Pi_{\mathcal{C}}(x) - x + t\big(\Pi_{\mathcal{C}}(y) - \Pi_{\mathcal{C}}(x)\big)\right\|^2
\end{aligned} \tag{A.1}$$

for all $t \in [0, 1]$. Since we have

$$\frac{\partial}{\partial t}\|\Pi_{\mathcal{C}}(x) - x + t(\Pi_{\mathcal{C}}(y) - \Pi_{\mathcal{C}}(x))\|^2 = 2\langle\Pi_{\mathcal{C}}(x) - x, \Pi_{\mathcal{C}}(y) - \Pi_{\mathcal{C}}(x)\rangle + 2t\|\Pi_{\mathcal{C}}(y) - \Pi_{\mathcal{C}}(x)\|^2 \tag{A.2}$$

and $\frac{\partial}{\partial t}\|\Pi_{\mathcal{C}}(x) - x + t(\Pi_{\mathcal{C}}(y) - \Pi_{\mathcal{C}}(x))\|^2 \ge 0$ at $t = 0$ by (A.1), we have

$$\langle\Pi_{\mathcal{C}}(x) - x, \Pi_{\mathcal{C}}(y) - \Pi_{\mathcal{C}}(x)\rangle \ge 0. \tag{A.3}$$

Likewise, we have

$$\langle\Pi_{\mathcal{C}}(y) - y, \Pi_{\mathcal{C}}(x) - \Pi_{\mathcal{C}}(y)\rangle \ge 0. \tag{A.4}$$

Now, consider the function

$$f(t) := \|\Pi_{\mathcal{C}}(x) - \Pi_{\mathcal{C}}(y) + t(x - \Pi_{\mathcal{C}}(x) - y + \Pi_{\mathcal{C}}(y))\|^2,$$

i.e. $f(0) = \|\Pi_{\mathcal{C}}(x) - \Pi_{\mathcal{C}}(y)\|^2$ and $f(1) = \|x - y\|^2$. Then $f(0) \le f(1)$ since the following inequality holds:

$$\begin{aligned}
\frac{d}{dt}f(t) &= 2\langle\Pi_{\mathcal{C}}(x) - \Pi_{\mathcal{C}}(y), x - \Pi_{\mathcal{C}}(x) - y + \Pi_{\mathcal{C}}(y)\rangle + 2t\|x - \Pi_{\mathcal{C}}(x) - y + \Pi_{\mathcal{C}}(y)\|^2 \\
&= 2\langle\Pi_{\mathcal{C}}(x) - x, \Pi_{\mathcal{C}}(y) - \Pi_{\mathcal{C}}(x)\rangle + 2\langle\Pi_{\mathcal{C}}(y) - y, \Pi_{\mathcal{C}}(x) - \Pi_{\mathcal{C}}(y)\rangle \\
&\quad + 2t\|x - \Pi_{\mathcal{C}}(x) - y + \Pi_{\mathcal{C}}(y)\|^2 \\
&\ge 0 \quad \forall t \in [0, 1]
\end{aligned}$$

where the inequality follows from (A.3) and (A.4). This completes the proof of Lemma A.1. $\quad\square$

**Lemma A.2** (Hoeffding's inequaltiy [Hoe63]). *Let $X$ be a random variable on $[a, b]$ and $X_1, \ldots, X_n$ be independent copies of $X$. Then, it holds that*

$$\mathbb{P}\left(\left|\frac{1}{n}\sum_{i=1}^{n} X_i - \mathbb{E}[X]\right| \ge \varepsilon\right) \le 2\exp\left(-\frac{2n\varepsilon^2}{(b-a)^2}\right).$$

**Lemma A.3.** *Suppose that $f : \mathbb{R}^d \to \mathbb{R}$ is convex and differentiable. Then, the following conditions are equivalent.*

1. *$f$ is $\beta$-smooth, i.e. $\|\nabla f(\theta) - \nabla f(\theta')\| \le \beta\|\theta - \theta'\|$ for all $\theta, \theta' \in \mathbb{R}^d$.*

2. *$\frac{\beta}{2}\|\theta\|^2 - f(\theta)$ is convex for all $\theta \in \mathbb{R}^d$.*

3. *$f(\theta) - f(\theta') - \nabla f(\theta')^\top(\theta - \theta') \le \frac{\beta}{2}\|\theta - \theta'\|^2$ for all $\theta, \theta' \in \mathbb{R}^d$.*

4. *$(\nabla f(\theta) - \nabla f(\theta'))^\top(\theta - \theta') \ge \frac{1}{\beta}\|\nabla f(\theta) - \nabla f(\theta')\|^2$ for all $\theta, \theta' \in \mathbb{R}^d$.*

*Proof.*

**1⇒2.** Let $g(\theta) := \frac{\beta}{2}\|\theta\|^2 - f(\theta)$. Then for any $\theta \neq \theta'$,

$$
\begin{aligned}
(\nabla g(\theta) - \nabla g(\theta'))^\top (\theta - \theta') &= \left(\beta(\theta - \theta') - (\nabla f(\theta) - \nabla f(\theta'))\right)^\top (\theta - \theta') \\
&= \beta\|\theta - \theta'\|^2 - (\nabla f(\theta) - \nabla f(\theta'))^\top (\theta - \theta') \\
&\geq \beta\|\theta - \theta'\|^2 - \|\nabla f(\theta) - \nabla f(\theta')\| \cdot \|\theta - \theta'\| \\
&\geq 0.
\end{aligned}
$$

In addition, by the mean value theorem, there exists $s \in (0,1)$ such that for $\theta_s = s\theta + (1-s)\theta'$,

$$
g(\theta') - g(\theta) = \nabla g(\theta_s)^\top (\theta' - \theta).
$$

Using this, we can derive the following inequality

$$
\begin{aligned}
0 &\leq (\nabla g(\theta_s) - \nabla g(\theta'))^\top (\theta_s - \theta') \\
&= s(\nabla g(\theta_s) - \nabla g(\theta'))^\top (\theta - \theta') \\
&= s(g(\theta) - g(\theta') - \nabla g(\theta')^\top (\theta - \theta')),
\end{aligned}
$$

which implies that $g$ is convex.

**2⇔3.** The following relation shows the equivalence of the second and the third statements:

$$
g(\theta) \geq g(\theta') + \nabla g(\theta')^\top (\theta - \theta') \iff f(\theta) - f(\theta') - \nabla f(\theta')^\top (\theta - \theta') \leq \frac{\beta}{2}\|\theta - \theta'\|^2.
$$

**2,3⇒4.** We first introduce the following claim.

**Claim 1.** *Suppose that $f : \mathbb{R}^d \to \mathbb{R}$ is convex and differentiable and have a global minimum $\theta^*$. Then, $\frac{1}{2\beta}\|\nabla f(\theta)\|^2 \leq f(\theta) - f(\theta^*)$.*

*Proof.* The statement of Claim 1 is a consequence of the following relation

$$
f(\theta^*) = \inf_{\theta'} f(\theta') \leq \inf_{\theta'} f(\theta) + \nabla f(\theta)^\top (\theta' - \theta) + \frac{\beta}{2}\|\theta' - \theta\|^2 = f(\theta) - \frac{1}{2\beta}\|\nabla f(\theta)\|^2.
$$

$\square$

Let $f_\theta(\phi) := f(\phi) - \nabla f(\theta)^\top \phi$. Since $\frac{\beta}{2}\|\phi\|^2 - f_\theta(\phi)$ is convex and $\phi = \theta$ minimizes $f_\theta$, from Claim 1, we have

$$
f(\theta') - f(\theta) - \nabla f(\theta)^\top (\theta' - \theta) = f_\theta(\theta') - f_\theta(\theta) \geq \frac{1}{2\beta}\|\nabla f(\theta) - \nabla f(\theta')\|^2,
$$

$$
f(\theta) - f(\theta') - \nabla f(\theta')^\top (\theta - \theta') = f_{\theta'}(\theta) - f_{\theta'}(\theta') \geq \frac{1}{2\beta}\|\nabla f(\theta) - \nabla f(\theta')\|^2.
$$

Adding two above inequalities derive the fourth statement.

**4⇒1.** The following inequality is sufficient for deriving the first statement

$$
\frac{1}{\beta}\|\nabla f(\theta) - \nabla f(\theta')\|^2 \leq (\nabla f(\theta) - \nabla f(\theta'))^\top (\theta - \theta') \leq \|\nabla f(\theta) - \nabla f(\theta')\| \cdot \|\theta - \theta'\|
$$

$$
\Rightarrow \|\nabla f(\theta) - \nabla f(\theta')\| \leq \beta\|\theta - \theta'\|.
$$

$\square$

## A.1 Contractivity of projected SGD for strongly convex and smooth objectives

**Lemma A.4.** *Let $f : \mathbb{R}^d \to \mathbb{R}$ be $\alpha$-strongly convex and $\beta$-smooth. Then for any $\eta \in (0, 2/\beta)$ and for any convex $\Theta \subset \mathbb{R}^d$, $\theta \mapsto \Pi_\Theta(\theta - \eta\nabla f(\theta))$ is $\sqrt{1 - 2\alpha\eta + \alpha\beta\eta^2}$-contractive.*

*Proof.* Let $g(\theta) := \theta - \eta \nabla f(\theta)$. Then for any $\theta, \theta' \in \mathbb{R}^d$,

$$\|g(\theta) - g(\theta')\|^2 = \|\theta - \theta'\|^2 - 2\eta(\nabla f(\theta) - \nabla f(\theta'))^\top (\theta - \theta') + \eta^2 \|\nabla f(\theta) - \nabla f(\theta')\|^2$$

$$\leq \|\theta - \theta'\|^2 - 2\eta \left( \left(1 - \frac{\beta\eta}{2}\right) \cdot \alpha \|\theta - \theta'\|^2 + \frac{\beta\eta}{2} \cdot \frac{1}{\beta} \|\nabla f(\theta) - \nabla f(\theta')\|^2 \right) + \eta^2 \|\nabla f(\theta) - \nabla f(\theta')\|^2$$

$$= (1 - 2\alpha\eta + \alpha\beta\eta^2)\|\theta - \theta'\|^2.$$

Here, the inequality is from the $\alpha$-strong convexity and Lemma A.3, i.e.

$$(\nabla f(\theta) - \nabla f(\theta'))^\top (\theta - \theta') \geq \alpha \|\theta - \theta'\|^2$$

$$(\nabla f(\theta) - \nabla f(\theta'))^\top (\theta - \theta') \geq \frac{1}{\beta} \|\nabla f(\theta) - \nabla f(\theta')\|^2.$$

Using Lemma A.1 completes the proof of Lemma A.4. $\qquad\qquad\square$

# B Proofs of results in Section 2

## B.1 Deriving generalization bounds using localized covers

In Lemma 2.1, we show that the union of all trajectories generated by $t \geq T = T_\varepsilon$ SGD iterations, i.e. $\bigcup_{\theta^{(0)} \in \Theta} \Psi_{\geq T}(\theta^{(0)})$, can be $\varepsilon$-covered by a set of points generated by exactly $T$ SGD iterations, i.e. $\Psi_T(0)$. Here, each $g_{i_T} \circ \cdots \circ g_{i_1}(0) \in \Psi_T(0)$ can be viewed as a *deterministic* algorithm mapping samples $z_1, \ldots, z_n$ to a parameter in $\Theta$, where each $g_{i_T} \circ \cdots \circ g_{i_1}(0)$ only depends on at most $T$ samples $z_{i_1}, \ldots, z_{i_T}$. Under this observation, in this section, we generalize our localized cover so that its elements are general deterministic algorithms depending on a small number of samples, i.e. not restricted to possible instances of SGD. Then, we derive generalization bounds using our (generalized) localized cover. To this end, we define algorithms depending on samples.

**Definition 5.** $\phi : \mathcal{Z}^n \to \Theta$ *is a "(deterministic) algorithm depending on at most $T$ samples" if there exists $\mathcal{I} \in \{\mathcal{S} \subset [n] : |\mathcal{S}| = T\}$ satisfying the following: for any $z_1, \ldots, z_n, z'_1, \ldots, z'_n \in \mathcal{Z}$ such that $z_i = z'_i$ for all $i \in \mathcal{I}$, $\phi(z_1, \ldots, z_n) = \phi(z'_1, \ldots, z'_n)$. Here, we refer to $\mathcal{I}$ as the "set of sample indices determining $\phi$." The collection of all algorithms depending on at most $T$ samples is denoted by $\mathcal{A}_T$.*

In Definition 5, we define $\phi$ to be an algorithm depending on at most $T$ samples if the value of $\phi$ can be fully determined by a subset of the samples $\{z_i\}_{i \in \mathcal{I}}$ for some $\mathcal{I} \in \{\mathcal{S} \subset [n] : |\mathcal{S}| = T\}$. We note that $\Psi_T(\theta^{(0)}) \subset \mathcal{A}_T$ for any $T \geq 0$ and $\theta^{(0)} \in \Theta$.

Now, we introduce the following theorem for deriving generalization bounds using a localized cover where each element in the cover is an algorithm depending on at most $T$ samples.

**Theorem B.1.** *Suppose that Assumptions 1 & 2 hold and $\Theta \subset \mathbb{R}^d$. Let $\varepsilon > 0$, $\Phi_n \subseteq \mathcal{A}_n$, and $\Phi_{T,\varepsilon} \subseteq \mathcal{A}_T$ such that*

$$\psi(x_{1:n}) \in \bigcup_{\phi \in \Phi_{T,\varepsilon}} \mathcal{B}_\varepsilon^d (\phi(x_{1:n})) \quad \text{for all } x_{1:n} \in \mathcal{Z}^n \text{ and } \psi \in \Phi_n. \tag{B.1}$$

*Let $\mu$ be a distribution over $\mathcal{Z}$ and $z_{1:n} = (z_1, \ldots, z_n)$ is such that $z_i$'s are i.i.d. samples from $\mu$. Then, with probability at least $1 - \delta$ over the sampling distribution of $z_{1:n}$, for any $\psi \in \Psi$,*

$$\left| \hat{F}(\psi(z_{1:n})) - F(\psi(z_{1:n})) \right| \leq \frac{BT}{n} + B\sqrt{\frac{\log(2|\Phi_{T,\varepsilon}|/\delta)}{2n}} + 2L\varepsilon.$$

Theorem B.1 implies that if (i) $\Phi_n$ can be covered by a set $\Phi_{T,\varepsilon}$ of algorithms depending on at most $T$ samples and (ii) $B$, $L$, $T$, and $|\Phi_{T,\varepsilon}|$ are independent of $d$, then a dimension-independent generalization bound can be derived. We note that the assumption (B.1) is a generalization of the observation in Lemma 2.1 since $\Psi_{\geq T} \subset \mathcal{A}_n$ and $\Psi_T \subset \mathcal{A}_T$.

*Proof of Theorem B.1.* From Assumption 1 and the assumption (B.1), for any $\psi \in \Phi_n$, we have

$$\left| \hat{F}(\psi(z_{1:n})) - F(\psi(z_{1:n})) \right| \leq \sup_{\phi \in \Phi_{T,\varepsilon}} \left| \hat{F}(\phi(z_{1:n})) - F(\phi(z_{1:n})) \right| + 2L\varepsilon. \tag{B.2}$$

Namely, if we bound $|\hat{F}(\phi(z_{1:n})) - F(\phi(z_{1:n}))|$ for all $\phi \in \Phi_{T,\varepsilon}$, the bound in Theorem B.1 follows. Since the statement of Theorem B.1 is trivial if $T = n$ or $|\Phi_{T,\varepsilon}| = \infty$, we assume $T < n$ and $|\Phi_{T,\varepsilon}| < \infty$. Now, we derive the target bound: for $\phi \in \Phi_{T,\varepsilon}$, $\theta := \phi(z_{1:n}) \in \Theta$, and the set of indices $\mathcal{I}_\phi$ determining $\phi$ with $|\mathcal{I}_\phi| \leq T$,

$$\left| \hat{F}(\theta) - F(\theta) \right| \leq \left| \frac{1}{n} \sum_{i \in \mathcal{I}_\phi} f(\theta; z_i) - F(\theta) \right| + \left| \frac{1}{n} \sum_{i \in [n] \setminus \mathcal{I}_\phi} f(\theta; z_i) - F(\theta) \right|$$

$$\leq \frac{B|\mathcal{I}_\phi|}{n} + \left| \frac{1}{n} \sum_{i \in [n] \setminus \mathcal{I}_\phi} f(\theta; z_i) - F(\theta) \right|$$

$$\leq \frac{B|\mathcal{I}_\phi|}{n} + \frac{n - |\mathcal{I}_\phi|}{n} \sqrt{\frac{B^2 \log(2|\Phi_{T,\varepsilon}|/\delta)}{2(n - |\mathcal{I}_\phi|)}} \quad \text{w.p.} \quad 1 - \delta/|\Phi_{T,\varepsilon}|$$

$$\leq \frac{BT}{n} + B\sqrt{\frac{\log(2|\Phi_{T,\varepsilon}|/\delta)}{2n}} \quad \text{w.p.} \quad 1 - \delta/|\Phi_{T,\varepsilon}|. \tag{B.3}$$

For the first inequality in (B.3), we use the triangle inequality to upper bound $|\hat{F}(\theta) - F(\theta)|$ using two terms: we only utilize the samples determining $\theta$ in the first term while remaining samples independent of $\theta$ are considered in the second term. The second inequality directly follows from Assumption 2. To bound the second term in RHS of the second inequality, one can apply the Hoeffding's inequality (see Lemma A.2), which leads us to the third inequality in (B.3). Here, the last inequality naturally follows. By using (B.2), (B.3), and by applying the union bound for all $\phi \in \Phi_{T,\varepsilon}$, we obtain the bound in Theorem B.1. $\qquad\square$

## B.2 Proof of Theorem 2.1

First, observe that $g_1, \ldots, g_n$ are $\gamma$-contractive by Lemma A.4. Let $\varepsilon = 1/(2Ln)$; then by Lemma 2.1, we have

$$\bigcup_{\theta^{(0)} \in \Theta} \Psi_{\geq T}(\theta^{(0)}) \subseteq \bigcup_{\phi \in \Psi_T(0)} \mathcal{B}_\varepsilon^d(\phi). \tag{B.4}$$

Now, we apply Theorem B.1 with

$$\Phi_n \leftarrow \bigcup_{\theta^{(0)} \in \Theta} \Psi_{\geq T}(\theta^{(0)}), \ \Phi_{T,\varepsilon} \leftarrow \Psi_T(0), \ \varepsilon \leftarrow \varepsilon, \ T \leftarrow T, \ \delta \leftarrow \delta.$$

where the assumption (B.1) in Theorem B.1 is satisfied by (B.4). This completes the proof of Theorem 2.1.

## B.3 Proof of Corollary 2.1

Let $\theta^{(t)} := g_{i_t} \circ \cdots \circ g_{i_1}(\theta^{(0)})$ and $\phi := g_{i_t} \circ \cdots \circ g_{t-T+1}(0)$, i.e. $\phi$ is an algorithm depending on at most $T$ samples. Since each $g_i$ is $\gamma$-contractive by Lemma A.4, one can observe that

$$\|\theta^{(t)} - \phi\| \leq \gamma^T R \leq \frac{1}{2Ln} =: \varepsilon. \tag{B.5}$$

Now, we apply Theorem B.1 with

$$\Phi_n \leftarrow \{\theta^{(t)}\}, \ \Phi_{T,\varepsilon} \leftarrow \{\phi\}, \ \varepsilon \leftarrow \varepsilon, \ T \leftarrow T, \ \delta \leftarrow \delta.$$

where the assumption (B.1) is satisfied by (B.5) and $|\Phi_\varepsilon| = 1$. This provides the bound in Corollary 2.1.

## B.4 Proof of Corollary 2.2

The proof of Corollary 2.2 is simple. Since $\theta^{(0)}$, $t$, and $i_1, \ldots, i_t$ are fixed, $\theta^{(t)} = g_{i_t} \circ \cdots \circ g_{i_1}(\theta^{(0)})$ is an algorithm depending on at most $t$ samples. Then by using Theorem B.1 with

$$\Phi_n, \Phi_{T,\varepsilon} \leftarrow \{\theta^{(t)}\}, \ \varepsilon \leftarrow 0, \ T \leftarrow t, \ \delta \leftarrow \delta,$$

we obtain the bound in Corollary 2.2.

## B.5 Derivation of (2.2)

We first formally define the Hausdorff dimension.

**Definition 6.** *Given $\mu \subset \mathbb{R}^d$, $d_H$ defined below is the "Hausdorff dimension" of $\mu$:*

$$d_H := \inf\{s \geq 0 : h^s(\mu) = 0\},$$

$$h^s(\mu) := \lim_{r \to 0} \inf \left\{ \sum_{i=1}^k r_i^s : k \in \mathbb{N} \cup \{\infty\}, (r_i)_{i=1}^k \in (0, r)^k \right.$$

$$\left. \textit{such that there exists } (\theta_i)_{i=1}^k \textit{ satisfying } \mu \subset \bigcup_i \mathcal{B}_{r_i}(\theta_i) \right\}.$$

From the assumption on $f(\theta; z)$, one can observe that for all $i \in [n]$ and $\theta, \theta' \in \mathcal{B}_R^d(0)$,

$$\|g_i(\theta) - g_i(\theta')\| \leq \gamma\|\theta - \theta'\|$$

where $\gamma = 1 - \eta$ since $\alpha = \beta = 1$. Then, the assumption $\|\theta_{z_i}^* - \theta_{z_j}^*\| \geq 2\gamma R$ guarantees that $g_i(\text{int}(\mathcal{B}_R)) \cap g_j(\text{int}(\mathcal{B}_R)) = \emptyset$ where $\text{int}(\mathcal{S})$ denotes the interior of a set $\mathcal{S}$.

Finally, the following theorem shows that $d_H = \frac{\log n}{\log(1/\gamma)}$. Substituting $d_H$ to the bound in Theorem 2.1 results in (2.2).

**Theorem B.2** (Theorem 9.3 in [Fal14]). *Suppose that $g_i(\text{int}(\mathcal{B}_R)) \cap g_j(\text{int}(\mathcal{B}_R)) = \emptyset$ for all $i \neq j$ and $\|g_i(\theta) - g_i(\theta')\| = \gamma\|\theta - \theta'\|$ for all $i$. Then $d_H = \frac{\log n}{\log(1/\gamma)}$.*

## B.6 Proof of Proposition 2.1

In this proof, we explicitly construct a piecewise $\beta$-strongly convex and $\beta$-smooth function, i.e. quadratic, which is always piecewise $\alpha$-strongly convex and $\beta$-smooth for any $\alpha \in [0, \beta]$. Let $\{\mathcal{Q}_1, \ldots, \mathcal{Q}_Q\}$ be the partition of $\mathcal{B}_R^d(0)$ such that $f = \ell_q$ on $\mathcal{Q}_q$ for some smooth $\ell_q$ on $\mathcal{B}_R^d(0)$. Given $q \in [Q]$ and $\varepsilon > 0$, let $\mathcal{C}_{q,\varepsilon} = \{\phi_{q,1}, \ldots, \phi_{q,P_q}\} \subset \Theta$ be an $\varepsilon$-cover of $\Theta$ with the minimum cardinality, i.e. $P_q \leq (3R/\varepsilon)^d$ for $\varepsilon \leq R$. Let $\{\mathcal{P}_{q,1}, \ldots, \mathcal{P}_{q,P_q}\}$ be a partition of $\mathcal{Q}_q$ where each $\mathcal{P}_{q,p}$ is defined as follows:

$$\mathcal{P}_{q,p} := \left\{ \theta \in \mathcal{Q}_q \setminus \bigcup_{r<p} \mathcal{P}_{q,r} : \|\phi_{q,p} - \theta\| \leq \min_{r>p} \|\phi_{q,r} - \theta\| \right\}.$$

For each $q \in [Q]$ and $p \in [P_q]$, we also define $h(\theta) := h_{q,p}(\theta)$ on $\mathcal{P}_{q,P_q}$ where

$$h_{q,p}(\theta) := \phi_{q,p} + \nabla\ell_q(\phi_{q,p})^\top(\theta - \phi_{q,p}) + \frac{\beta}{2}\|\theta - \phi_{q,p}\|^2$$

and $\nabla_j$ denotes the partial derivative with respect to the $j$-th entry. Then, one can observe that $h$ is piecewise $\alpha$-strongly convex and $\beta$-smooth with $Q(3R/\varepsilon)^d$ pieces for any $\alpha \leq \beta$. Furthermore, for any $\theta \in \mathcal{P}_{q,p}$, we have

$$\|\nabla f(\theta) - \nabla h(\theta)\| = \|\nabla\ell_q(\theta) - \nabla\ell_q(\phi_{q,p}) - \beta(\theta - \phi_{q,p})\| \leq (\beta + \beta')\varepsilon.$$

Choosing $\varepsilon := \xi/(\beta + \beta')$ completes the proof of Proposition 2.1.

## B.7 Proof of Theorem 2.2

Given $z \in \mathcal{Z}$, let $\mathcal{P}_{z,1}, \ldots, \mathcal{P}_{z,P}$ be a partition of $\Theta$ and $h_1(\cdot; z), \ldots, h_P(\cdot; z)$ be $\alpha$-strongly convex and $\beta$-smooth functions such that $h(\cdot; z) = h_p(\cdot; z)$ on $\mathcal{P}_{z,p}$ for all $p \in [P]$. Let $g_{i,p}(\theta) := \theta - \eta\nabla h_p(\theta; z_i)$, i.e. each $g_{i,p}$ is $\gamma$-contractive by Lemma A.4. Given $\psi^{(0)} \in \Theta$, $t \geq T$, and $i_1, \ldots, i_t \in [n]$, let $\psi^{(s)} := g_{i_s} \circ \cdots \circ g_{i_1}(\psi^{(0)})$ for all $s \in [t]$ and let $p_s \in [P]$ be an index satisfying $\psi^{(s-1)} \in \mathcal{P}_{z_{i_s}, p_s}$. Let $\phi := g_{i_t, p_t} \circ \cdots \circ g_{i_{t-T+1}, p_{t-T+1}}(0)$, i.e. $\phi$ is an algorithm depending on at most $T$ samples. Then for any $\theta \in \Theta$, we have

$$\begin{aligned}
\|\psi^{(s)} - g_{i_s, p_s}(\theta)\| &= \|\psi^{(s-1)} - \eta\nabla f(\psi^{(s-1)}; z_{i_s}) - g_{i_s, p_s}(\theta)\| \\
&= \|\psi^{(s-1)} - \eta\nabla h(\psi^{(s-1)}, z_{i_s}) + \eta\nabla h(\psi^{(s-1)}, z_{i_s}) - \eta\nabla f(\psi^{(s-1)}; z_{i_s}) - g_{i_s, p_s}(\theta)\| \\
&\leq \|g_{i_s, p_s}(\psi^{(s-1)}) - g_{i_s, p_s}(\theta)\| + \eta\|\nabla f(\psi^{(s-1)}; z_{i_s}) - \nabla h(\psi^{(s-1)}; z_{i_s})\| \\
&\leq \gamma\|\psi^{(s-1)} - \theta\| + \eta\xi
\end{aligned}$$

This implies that

$$\|\psi^{(t)} - \phi\| \leq \gamma^T\|\psi^{(0)}\| + \eta\xi\sum_{t=0}^{T-1}\gamma^t \leq \gamma^T R + \frac{1-\gamma^T}{1-\gamma}\eta\xi =: \varepsilon. \tag{B.6}$$

Now, we apply Theorem B.1 with

$$\Phi_n \leftarrow \bigcup_{\theta^{(0)} \in \Theta} \Psi_{\geq T}(\theta^{(0)}), \; \Phi_{T,\varepsilon} \leftarrow \{g_{i_T, p_T} \circ \cdots \circ g_{i_1, p_1}(0) : i_1, \ldots, i_T \in [n], p_1, \ldots, p_T \in [P]\},$$

$$\varepsilon \leftarrow \varepsilon, \; T \leftarrow T, \; \delta \leftarrow \delta$$

where the assumption (B.1) in Theorem B.1 is satisfied by (B.6) and $|\Phi_{T,\varepsilon}| \leq (nP)^T$. This leads us to the bound in Theorem 2.2.

# C   Proofs of results in Section 3

## C.1   Proof of Theorem 3.1

Given $z = (y, x) \in \mathcal{Y} \times \mathcal{B}^d_{R_x}(0) = \mathcal{Z}$, let $\{\mathcal{Q}_{z,1}, \ldots, \mathcal{Q}_{z,Q}\}$ be a partition of $\Theta$ such that $\ell(\cdot, \ldots, \cdot; y)$ is $\beta$-smooth on $\mathcal{Q}_{z,q}$ for all $q \in [Q]$. For proving Theorem 3.1, we utilize Theorem 2.2 by approximating each $f(\cdot; z)$ using a piecewise strongly convex and smooth function $h(\cdot; z)$ with $P^K Q$ pieces. To this end, we first define $P$ by

$$P := \left\lceil \frac{2RR_x}{\kappa} \right\rceil \quad \text{where} \quad \kappa := \frac{1}{12\beta\eta K L R_x T n}.$$

Using $P$ defined as above, for each $z = (y, x) \in \mathcal{Y} \times \mathcal{B}^d_{R_x}(0)$, we construct a partition $\{\mathcal{P}_{z,q,p_1,\ldots,p_K} : p_1, \ldots, p_K \in [P], q \in [Q]\}$ of $\Theta$ as follows

$$\mathcal{P}_{z,q,p_1,\ldots,p_K} := \{(\theta_j)^K_{j=1} : (\theta_1^\top x, \ldots, \theta_K^\top x) \in \mathcal{Q}_{z,q}, \theta_j^\top x \in \mathcal{T}_{p_j} \; \forall j \in [K]\}$$

where

$$\mathcal{T}_p := \begin{cases} [\mu_p, \mu_{p+1}) & \text{if } p \in [P-1] \\ [\mu_P, \mu_{P+1}] & \text{if } p = P \end{cases},$$

$$\mu_p := -RR_x + (p-1)\kappa \quad \forall p \in [P+1].$$

For the notational simplicity, let $u := (z, q, p_1, \ldots, p_K) \in \mathcal{Z} \times [Q] \times [P]^K$. Now, we define our approximation as $h(\cdot; z) = h_u(\cdot)$ on $\mathcal{P}_u$ if $\mathcal{P}_u \neq \emptyset$. Here, $h_u(\cdot)$ is defined as follows: for some fixed $\nu_u := (\nu_{u,1}, \ldots, \nu_{u,K}) \in \mathcal{P}_u$,

$$h_u(\theta) := \ell(\nu_{u,1}^\top x, \ldots, \nu_{u,K}^\top x; y) + \sum_{j=1}^K \nabla_j \ell(\nu_{u,1}^\top x, \ldots, \nu_{u,K}^\top x; y)(\theta_j^\top x - \nu_{u,j}^\top x) + \sum_{j=1}^K \frac{\lambda}{2}\|\theta_j\|^2.$$

where $\nabla_j$ denotes the partial derivative with respect to the $j$-th argument. Namely, $h_u$ is a first order approximation of $\ell$ at $\theta$ with $\ell_2$-regularization; hence, $h_u$ is $\lambda$-strongly convex and $\lambda$-smooth. Then given $z \in \mathcal{Z}$ and $\theta \in \Theta$, for $q$ and $p_1, \ldots, p_K$ satisfying $\theta \in \mathcal{P}_u$ for $u = (z, q, p_1, \ldots, p_K)$, one can observe that

$$\|\nabla f(\theta; z) - \nabla h_u(\theta)\| \leq K^{1/2} \max_{j \in [K]} \|\nabla_j \ell(\theta_1^\top x, \ldots, \theta_K^\top x; y)x - \nabla_j \ell(\nu_{u,1}^\top x, \ldots, \nu_{u,K}^\top x; y)x\|$$

$$\leq \beta\kappa K R_x = \frac{1}{12\eta L T n}.$$

Now, we apply Theorem 2.2 with

$$h \leftarrow h, \; P \leftarrow P^K Q, \; \xi \leftarrow \frac{1}{12\eta L T n}, \; T \leftarrow T, \; \alpha \leftarrow \lambda, \; \beta \leftarrow \lambda, \; L \leftarrow L, \; B \leftarrow B.$$

Then in the bound of Theorem 2.2, we have

$$2L\left(\gamma^T R + \frac{1 - \gamma^T}{1 - \gamma}\eta\xi\right) \leq \frac{1}{n}$$

since $\gamma^T R \leq 1/(3Ln)$ and $\frac{1-\gamma^T}{1-\gamma} = \sum_{j=0}^{T-1} \gamma^j \leq T$. This completes the proof of Theorem 3.1.

## C.2   Equivalence between soft $K$-means algorithm and expectation-maximization for (3.4)

We first observe that applying the affine transformation $x \mapsto -nKx + \log(\frac{(\xi/\pi)^{d/2}}{K})$ to (3.4) results in the following objective:

$$\sum_{i=1}^n \log\left(\frac{1}{K}(\zeta/\pi)^{d/2} \sum_{j=1}^K \exp\left(-\zeta\|\theta_j - z_i\|^2\right)\right) \tag{C.1}$$

i.e. the expectation-maximization algorithm for (C.1) aims to find a local minimum of (3.4). Since (C.1) is the log-likelihood for the mixture of Gaussians $N(\theta_1, \frac{1}{2\zeta}I), \ldots, N(\theta_K, \frac{1}{2\zeta}I)$ under the same cluster density $1/K$ with observations $z_1, \ldots, z_n$, the expectation-maximization algorithm for (C.1) is identical to the alternative procedure (3.3) [Pri12].

## C.3  Proof of Theorem 3.2

First, observe that for $\theta \in (\mathcal{B}_R^d(0))^K, z \in \mathcal{B}_R^d(0)$, we have

$$\frac{\partial}{\partial \theta_k} f(\theta; z) = \frac{2(\theta_k - z_i) \exp(-\zeta \|\theta_k - z_i\|^2)}{\sum_{j=1}^K \exp(-\zeta \|\theta_j - z_i\|^2)},$$

$$\frac{\partial^2}{\partial \theta_k \partial \theta_{k'}} f(\theta; z) = \frac{4\zeta(\theta_k - z_i)(\theta_{k'} - z_i)^\top \exp(-\zeta \|\theta_k - z_i\|^2 - \zeta \|\theta_{k'} - z_i\|^2)}{\left(\sum_{j=1}^K \exp(-\zeta \|\theta_j - z_i\|^2)\right)^2}$$

$$+ \mathbf{1}_{k=k'} \times \frac{(2I - 4\zeta(\theta_k - z_i)(\theta_k - z_i)^\top) \exp(-\zeta \|\theta_k - z_i\|^2)}{\sum_{j=1}^K \exp(-\zeta \|\theta_j - z_i\|^2)}$$

where $\mathbf{1}_{k=k'}$ is one if $k = k'$ and zero otherwise. Using this, we bound $f$ and its first and second derivatives as follows:

$$f(\theta; z) \in [-B + \log K, \log K],$$

$$L := \frac{4R}{\sqrt{K}} \exp(\zeta B) = 4R \sqrt{\frac{K}{K^2 \exp(-\zeta B)^2}} \geq 4R \sqrt{\frac{\sum_{j=1}^K \exp(-\zeta \|\theta_j - z_i\|^2)^2}{\left(\sum_{j=1}^K \exp(-\zeta \|\theta_j - z_i\|^2)\right)^2}}$$

$$\geq \sqrt{\sum_{k=1}^K \left\| \frac{\partial}{\partial \theta_k} f(\theta; z) \right\|^2} = \left\| \frac{\partial}{\partial \theta} f(\theta; z) \right\|,$$

$$\alpha := \frac{2}{K} \exp(-\zeta B) \leq \frac{1}{\|\theta_j - z\|} \left\| \frac{\partial}{\partial \theta_j} f(\theta; z) \right\| \quad \forall \theta \text{ s.t. } \theta_j \neq z,$$

$$\beta := \frac{2}{K} \exp(\zeta B) \geq \frac{1}{\|\theta_j - z\|} \left\| \frac{\partial}{\partial \theta_j} f(\theta; z) \right\| \quad \forall \theta \text{ s.t. } \theta_j \neq z,$$

$$\beta' := 4\zeta B \exp(\zeta B) + 4\zeta B + 2 \geq \left\| \nabla^2 f(\theta; z) \right\|_2.$$

To utilize Theorem 2.2, we approximate each $f(\cdot; z)$ using a piecewise strongly convex and smooth function $h(\cdot; z)$ with $P^K$ pieces. To this end, we first construct a partition $\{\mathcal{P}_{z,p_1,\ldots,p_K} : p_1, \ldots, p_K \in [P]\}$ of $\Theta$ for each $z \in \mathcal{Z}$ as follows:

$$\mathcal{P}_{z,p_1,\ldots,p_K} := \{(\theta_j)_{j=1}^K \in \Theta : \|\theta_j - z\| \in \mathcal{T}_{p_j} \ \forall j \in [K]\}$$

where

$$\kappa := \frac{1}{12(\beta + \beta')\eta \sqrt{K} L T n},$$

$$P := \left\lceil \frac{2R}{\kappa} \right\rceil,$$

$$\mu_p := (p-1)\kappa \quad \forall p \in [P+1],$$

$$\mathcal{T}_p := \begin{cases} [\mu_p, \mu_{p+1}) & \text{if } p \in [P-1] \\ [\mu_P, \mu_{P+1}] & \text{if } p = P \end{cases}.$$

We define $h(\cdot; z) := h_{z,p_1,\ldots,p_K}(\cdot)$ on $\mathcal{P}_{z,p_1,\ldots,p_K}$ if $\mathcal{P}_{z,p_1,\ldots,p_K} \neq \emptyset$. Here, note that $\mathcal{P}_{z,p_1,\ldots,p_K} \setminus \{z\} \neq \emptyset$ if $\mathcal{P}_{z,p_1,\ldots,p_K} \neq \emptyset$ from the definition of $\mathcal{P}_{z,p_1,\ldots,p_K}$. Now, we define $h_{z,p_1,\ldots,p_K}(\cdot)$ as follows: for some fixed $\nu_{z,p_1,\ldots,p_K} = ((\nu_{z,p_1,\ldots,p_K})_j)_{j=1}^K \in \mathcal{P}_{z,p_1,\ldots,p_K} \setminus \{z\}$,

$$h_{z,p_1,\ldots,p_K}(\theta) := \sum_{j=1}^K \frac{1}{2} \left\| \frac{1}{\|(\nu_{z,p_1,\ldots,p_K})_j - z\|} \frac{\partial}{\partial \theta_j'} f(\theta'; z) \Big|_{\theta' = \nu_{z,p_1,\ldots,p_K}} \right\| \cdot \|\theta_j - z\|^2.$$

Our construction of $h_{z,p_1,\ldots,p_K}$ has some nice properties. For example, for

$$a_{z,p_1,\ldots,p_K} := \min_j \left\| \frac{1}{\|(\nu_{z,p_1,\ldots,p_K})_j - z\|} \frac{\partial}{\partial \theta_j'} f(\theta'; z) \Big|_{\theta' = \nu_{z,p_1,\ldots,p_K}} \right\|,$$

$$b_{z,p_1,\ldots,p_K} := \max_j \left\| \frac{1}{\|(\nu_{z,p_1,\ldots,p_K})_j - z\|} \frac{\partial}{\partial \theta_j'} f(\theta'; z) \Big|_{\theta' = \nu_{z,p_1,\ldots,p_K}} \right\|,$$

$h_{z,p_1,\ldots,p_K}$ is $a_{z,p_1,\ldots,p_K}$-strongly convex and $b_{z,p_1,\ldots,p_K}$-smooth and $\alpha \leq a_{z,p_1,\ldots,p_K} \leq b_{z,p_1,\ldots,p_K} \leq \beta$. Furthermore, for any $\nu' = (\nu'_j)_{j=1}^K$ satisfying

$$\|\nu'_j - z\| = \|(\nu_{z,p_1,\ldots,p_K})_j - z\|, \tag{C.2}$$

we have

$$\nabla f(\nu', z) = \nabla h_{z,p_1,\ldots,p_K}(\nu') \tag{C.3}$$

from the symmetry of $f$ and $h_{z,p_1,\ldots,p_K}$.

Now, we bound $\|\nabla f(\theta; z) - \nabla f(\theta; z)\|$ to utilize Theorem 2.2. For $\theta = (\theta_j)_{j=1}^K \in (\mathcal{B}_R^d(0))^K$, let $p_1,\ldots,p_K$ be indices in $[P]$ satisfying $\theta \in \mathcal{P}_{z,p_1,\ldots,p_K}$. Let $\nu'$ be a point on the line connecting $z$ and $\theta$ such that $\nu'$ satisfies (C.2), i.e. $\|\nu'_j - \theta_j\| \leq \kappa$ for all $j \in [K]$. Then, we have

$$\|\nabla f(\theta; z) - \nabla h(\theta; z)\| = \|\nabla f(\theta; z) - \nabla f(\nu'; z) + \nabla f(\nu'; z) - \nabla h(\theta; z)\|$$
$$\leq \|\nabla f(\theta; z) - \nabla f(\nu'; z)\| + \|\nabla h(\theta; z) - \nabla h_{z,p_1,\ldots,p_K}(\nu')\|$$
$$\leq \beta'\kappa\sqrt{K} + \sqrt{K} \max_{j \in [K]} \left\{ \left\| \frac{\partial}{\partial \theta_j} h_{z,p_1,\ldots,p_K}(\theta) - \frac{\partial}{\partial \theta_j} h_{z,p_1,\ldots,p_K}(\nu') \right\| \right\}$$
$$\leq (\beta + \beta')\kappa\sqrt{K}$$
$$\leq \frac{1}{12\eta LTn}$$

The first inequality holds since $\nabla f(\nu'; z) = \nabla h_{z,p_1,\ldots,p_K}(\nu')$ by (C.3) and other inequalities holds from the definitions of $\beta$ and $\beta'$. Now, we apply Theorem 2.2 with

$$h \leftarrow h, \ P \leftarrow P^K, \ \xi \leftarrow \frac{1}{12\eta LTn}, \ T \leftarrow T, \ \alpha \leftarrow \alpha, \ \beta \leftarrow \beta, \ L \leftarrow L, \ B \leftarrow B.$$

Then in the bound of Theorem 2.2, we have

$$2L\left(\gamma^T R + \frac{1-\gamma^T}{1-\gamma}\eta\xi\right) \leq \frac{1}{n}$$

since $\gamma^T R \leq 1/(3Ln)$ and $\frac{1-\gamma^T}{1-\gamma} = \sum_{j=0}^{T-1} \gamma^j \leq T$. This completes the proof of Theorem 3.2.

## C.4 Proof of Theorem 3.3

For $i \in [n]$ and $j \in [K]$, we first define $g_{i,j}$ as

$$g_{i,j}(\theta) := (\theta'_k)_{k=1}^K \quad \text{where} \quad \theta'_k = \begin{cases} \theta_k - 2\eta(\theta_k - z_i) & \text{if } k = j \\ \theta_k & \text{if } k \neq j \end{cases}.$$

For $\varepsilon := \frac{1}{8Rn}$, We define

$$\Phi_\varepsilon := \{(\theta'_j)_{j=1}^K : \theta'_j = \left(g_{i_{j,t_j},j} \circ \cdots \circ g_{i_{j,1},j}(\theta^{(0)})\right)_j, t_j \in [T] \cup \{0\}, i_{j,1},\ldots,i_{j,t_j} \in [n], \forall j \in [K]\}.$$

Given $\theta^{(0)}$, since $g_{i,j}$ is $\gamma$-contractive by Lemma A.4, we have for any $t \geq 0$, $i_1,\ldots,i_t \in [n]$, and $j \in [K]$,

$$\left(g_{i_t} \circ \cdots \circ g_{i_1}(\theta^{(0)})\right)_j \in \bigcup_{\phi \in \Phi_\varepsilon} \mathcal{B}_{K^{-1/2}\varepsilon}^d(\phi_j),$$

i.e.

$$g_{i_t} \circ \cdots \circ g_{i_1}(\theta^{(0)}) \in \bigcup_{\phi \in \Phi_\varepsilon} \mathcal{B}_\varepsilon^d(\phi), \tag{C.4}$$

Namely, $\Phi_\varepsilon$ contains all possible SGD parameters starting from $\theta^{(0)}$ where each element in $\Phi_\varepsilon$ is an algorithm depending on at most $KT$ samples. Since each $f(\cdot; z)$ is $(4R)$-Lipschitz on $\Theta$, we apply Theorem B.1 with

$$\Phi_n \leftarrow \{g_{i_t} \circ \cdots \circ g_{i_1}(\theta^{(0)}) : t \geq 0, i_1,\ldots,i_t \in [n]\}$$
$$\Phi_{T,\varepsilon} \leftarrow \Phi_\varepsilon, \ \varepsilon \leftarrow \varepsilon, \ T \leftarrow KT, \ \delta \leftarrow \delta, \ L \leftarrow 4R, \ B \leftarrow B.$$

where

$$|\Phi_\varepsilon| \leq \sum_{t_1,\ldots,t_K \in [T] \cup \{0\}} n^{\sum_{k=1}^K t_k} \leq \sum_{t_1,\ldots,t_K \in [T] \cup \{0\}} n^{KT} \leq (T+1)^K \cdot n^{KT} \leq (2n)^{KT}$$

and the assumption (B.1) in Theorem B.1 is satisfied by (C.4). This leads us to the bound in Theorem 3.3.

# D  Expected generalization gap: Strongly convex and smooth case

In this section, we bound the expected generalization gap using our localized cover for a contractive iterative stochastic optimizer $g$ (see Definition 3), i.e. each $g(\cdot;z)$ is contractive for all $z \in \mathcal{Z}$. Here, we use $\mathcal{S}$ for denoting the set of samples $\{z_1, \ldots, z_n\}$ and $g_i$ for denoting $g(\cdot;z_i)$.

**Theorem D.1.** *Suppose that Assumptions 1 & 2 hold and there exists $\gamma \in (0,1)$ such that $g(\cdot;z)$ is $\gamma$-contractive on $\Theta \subseteq \mathcal{B}_R^d(0)$ for all $z \in \mathcal{Z}$. Let $T := \max\left\{ \left\lceil \frac{\log(2LRn)}{\log(1/\gamma)} \right\rceil, 0 \right\}$. Then for any $\theta^{(0)} \in \Theta$, $t \geq 0$, and $i_1, \ldots, i_t \in [n]$, the $t$-th iterate $\theta^{(t)} := g_{i_t} \circ \cdots \circ g_{i_1}(\theta^{(0)})$ satisfies*

$$|\mathbb{E}_{\mathcal{S}}[\hat{F}(\theta^{(t)}) - F(\theta^{(t)})]| \leq \frac{BT + 1}{n}.$$

Compared to the uniform stability-based bound for $\gamma$-contractive $\frac{1}{n}\sum_{i=1}^n g_i$ under $L'$-Lipschitz $f(\cdot;z)$ and $\|g(\theta;z)\| \leq K$ [KWS22]

$$|\mathbb{E}_{\theta^{(0)}, i_1, \ldots, i_t, \mathcal{S}}[\hat{F}(\theta^{(t)}) - F(\theta^{(t)})]| \leq \frac{2KL'}{(1-\gamma)n}, \tag{D.1}$$

our bound in Theorem D.1 is linear in $B$, has an additional $\log n$ factor, and requires contractive $g(\cdot;z)$ but does not depend on $L^\dagger$ and $K$. Here, note that our bound uniformly holds for all parameters generated by SGD while the bound (D.1) requires the expectation over $\theta^{(0)}$ and the indices $i_1, \ldots, i_t$ used for the $t$ updates using the iterative stochastic optimizer.

Although the bounds in Theorem D.1 and (D.1) provide some information about the expected generalization gap, these bounds cannot be used for understanding the absolute deviation $|\hat{F}(\theta^{(t)}) - F(\theta^{(t)})|$, which can be of practical interest. In the following theorem, we provide a bound on the expectation of the absolute generalization gap where $\Psi_{\geq T}(\theta)$ and $\Psi_T(\theta)$ are defined for general $g_i$, analogous to Definition 1:

$$\Psi_T(\theta^{(0)}) := \{g_{i_T} \circ \cdots \circ g_{i_1}(\theta^{(0)}) : i_1, \ldots, i_T \in [n]\},$$
$$\Psi_{\geq T}(\theta^{(0)}) := \bigcup_{t \geq T} \Psi_t(\theta^{(0)}). \tag{D.2}$$

**Theorem D.2.** *Assume the setup in Theorem D.1. Then there exists some absolute constant $C > 0$ such that*

$$\mathbb{E}_{\mathcal{S}}\left[ \sup_{\theta^{(t)} \in \bigcup_{\theta^{(0)} \in \Theta} \Psi_{\geq T}(\theta^{(0)})} |\hat{F}(\theta^{(t)}) - F(\theta^{(t)})| \right] \leq \frac{BT + 1}{n} + CB\sqrt{\frac{T \log n}{n}}.$$

To our knowledge, the bound in Theorem D.2 is the first bound on $\mathbb{E}_{\mathcal{S}}[|\hat{F}(\theta^{(t)}) - F(\theta^{(t)})|]$ for the contractive case, including strongly convex and smooth functions. Compared to Theorem D.1, the bound in Theorem D.2 looses $O(\sqrt{T/(n\log n)})$ factor for taking the absolute value inside the expectation. Nevertheless, this bound can be improved by removing the supremum inside the expectation.

**Corollary D.1.** *Assume the setup in Theorem 2.1. Then there exists some absolute constant $C > 0$ such that for any $\theta^{(0)} \in \Theta$, $t \geq 0$, and $i_1, \ldots, i_t \in [n]$, we have*

$$\mathbb{E}_{\mathcal{S}}\left[ |\hat{F}(\theta^{(t)}) - F(\theta^{(t)})| \right] \leq \frac{BT + 1}{n} + CB\sqrt{\frac{1}{n}}.$$

Compared to Theorem D.2, the bound in Corollary D.1 does not have $\sqrt{T \log n}$ factor but the LHS in the bound is weaker.

## D.1  Proof of Theorem D.1

For any $\theta^{(t)} \in \bigcup_{\theta^{(0)} \in \Theta} \Psi_{\geq 0}(\theta^{(0)})$, let $\theta^{(t)} = g_{i_t} \circ \cdots \circ g_{i_1}(\theta^{(0)})$. Let $\phi = g_{i_t} \circ \cdots \circ g_{i_{t-T+1}}(0)$ and $\mathcal{I} = \{i_{t-T+1}, \ldots, i_t\} \subset [n]$ if $t > T$ and $\phi = g_{i_t} \circ \cdots \circ g_{i_1}(0)$ and $\mathcal{I} = \{i_1, \ldots, i_t\} \subset [n]$ otherwise. Then as in the proof of Lemma 2.1, we have

$$\theta^{(t)} \in \mathcal{B}_\varepsilon^d(\phi) \tag{D.3}$$

for $\varepsilon = 1/(2Ln)$, regardless of the choice of $\mathcal{S}$. Using this, we have

$$|\mathbb{E}_{\mathcal{S}}[\hat{F}(\theta^{(t)}) - F(\theta^{(t)})]| \le |\mathbb{E}_{\mathcal{S}}[(\hat{F}(\theta^{(t)}) - \hat{F}(\phi)]| + |\mathbb{E}_{\mathcal{S}}[F(\theta^{(t)}) - F(\phi)]| + |\mathbb{E}_{\mathcal{S}}[\hat{F}(\phi) - F(\phi)]|$$

$$\le 2L\varepsilon + |\mathbb{E}_{\mathcal{S}}[\hat{F}(\phi) - F(\phi)]|$$

$$\le 2L\varepsilon + \left|\mathbb{E}_{\mathcal{S}}\left[\frac{1}{n}\sum_{i\in[n]\setminus\mathcal{I}}(\hat{f}(\phi; z_i) - F(\phi))\right]\right| + \left|\mathbb{E}_{z_i:i\in\mathcal{I}}\left[\frac{1}{n}\sum_{i\in\mathcal{I}}(\hat{f}(\phi; z_i) - F(\phi))\right]\right|$$

$$= 2L\varepsilon + \left|\mathbb{E}_{z_i:i\in\mathcal{I}}\left[\frac{1}{n}\sum_{i\in\mathcal{I}}(\hat{f}(\phi; z_i) - F(\phi))\right]\right|$$

$$\le 2L\varepsilon + \frac{BT}{n} = \frac{BT+1}{n}.$$

The first inequality is from the triangle inequality and the second inequality is from (D.3). The third inequality is again from the triangle inequality and the first equality is from the fact that $\phi$ is independent of $\{z_i : i \in \mathcal{I}\}$. The last inequality is from Assumption 2. Since the above bound holds for any $\theta^{(0)} \in \Theta$ and $\mathcal{I}_t$, this completes the proof of Theorem D.1.

## D.2 Proof of Theorem D.2

In this proof, we assume $T < n$ since the statement trivially follows otherwise. As in the statement of Lemma 2.1, one can observe that

$$\bigcup_{\theta^{(0)}\in\Theta} \Psi_{\ge T}(\theta^{(0)}) \subseteq \bigcup_{\phi\in\Psi_T(0)} \mathcal{B}_\varepsilon^d(\phi) \tag{D.4}$$

for $\varepsilon = 1/(2Ln)$. For each $\phi \in \Psi_T(0)$, let $\mathcal{I}_\phi := \{i_1, \ldots, i_T\}$ such that $\phi = g_{i_T} \circ \cdots \circ g_{i_1}(0)$. Using this we can derive the following bound: for $\Pi_{\Psi_T(0)}(\theta^{(t)}) := \arg\min_{\phi\in\Psi_T(0)} \|\theta^{(t)} - \phi\|$,

$$\mathbb{E}_{\mathcal{S}}\left[\sup_{\theta^{(t)}\in\bigcup_{\theta^{(0)}\in\Theta}\Psi_{\ge T}(\theta^{(0)})}|\hat{F}(\theta^{(t)}) - F(\theta^{(t)})|\right] \le \mathbb{E}_{\mathcal{S}}\left[\sup_{\phi\in\Psi_T(0)}|\hat{F}(\phi) - F(\phi)|\right]$$

$$+ \mathbb{E}_{\mathcal{S}}\left[\sup_{\theta^{(t)}\in\bigcup_{\theta^{(0)}\in\Theta}\Psi_{\ge T}(\theta^{(0)})}|\hat{F}(\theta^{(t)}) - \hat{F}(\Pi_{\Psi_T(0)}(\theta^{(t)}))| + |F(\theta^{(t)}) - F(\Pi_{\Psi_T(0)}(\theta^{(t)}))|\right]$$

$$\le 2L\varepsilon + \mathbb{E}_{\mathcal{S}}\left[\sup_{\phi\in\Psi_T(0)}|\hat{F}(\phi) - F(\phi)|\right]$$

$$\le \frac{1}{n} + \mathbb{E}_{\mathcal{S}}\left[\sup_{\phi\in\Psi_T(0)}\left|\frac{1}{n}\sum_{i\in\mathcal{I}_\phi}(f(\phi; z_i) - F(\phi))\right|\right] + \mathbb{E}_{\mathcal{S}}\left[\sup_{\phi\in\Psi_T(0)}\left|\frac{1}{n}\sum_{i\in[n]\setminus\mathcal{I}_\phi}(f(\phi; z_i) - F(\phi))\right|\right]$$

$$\le \frac{BT+1}{n} + \mathbb{E}_{\mathcal{S}}\left[\sup_{\phi\in\Psi_T(0)}\left|\frac{|[n]\setminus\mathcal{I}_\phi|}{n}\cdot\frac{1}{|[n]\setminus\mathcal{I}_\phi|}\sum_{i\in[n]\setminus\mathcal{I}_\phi}(f(\phi; z_i) - F(\phi))\right|\right]$$

$$\le \frac{BT+1}{n} + CB\sqrt{\frac{T\log n}{n}}.$$

The first inequality is from the triangle inequality and the second inequality is from (D.4). The third inequality is again from the triangle inequality and the definition of $\varepsilon$, and the fourth inequality is from Assumption 2. Since each $\frac{1}{|[n]\setminus\mathcal{I}_\phi|}\sum_{i\in[n]\setminus\mathcal{I}_\phi}(f(\phi; z_i) - F(\phi))$ is sub-Gaussian with the sub-Gaussian norm bounded by $cB/\sqrt{|[n]\setminus\mathcal{I}_\phi|}$ for some absolute constant $c$ (see Proposition 2.5.2 in [Ver18]), the last inequality follows from a standard upper bound for empirical processes (see Exercise 2.5.10 in [Ver18]). This completes the proof of Theorem D.2.

## D.3 Proof of Corollary D.1

As in the proof of Theorem D.2, we assume that $T < n$ without loss of generality. The proof here is almost identical to that of Theorems D.1 & D.2.

For any $\theta^{(t)} \in \bigcup_{\theta^{(0)} \in \Theta} \Psi_{\geq 0}(\theta^{(0)})$, let $\theta^{(t)} = g_{i_t} \circ \cdots \circ g_{i_1}(\theta^{(0)})$. Let $\phi = g_{i_t} \circ \cdots \circ g_{i_{t-T+1}}(0)$ and $\mathcal{I} = \{i_{t-T+1}, \ldots, i_t\} \subset [n]$ if $t > T$ and $\phi = g_{i_t} \circ \cdots \circ g_{i_1}(0)$ and $\mathcal{I} = \{i_1, \ldots, i_t\} \subset [n]$ otherwise. Then for $\varepsilon = 1/(2Ln)$, we have

$$\theta^{(t)} \in \mathcal{B}_\varepsilon(\phi). \tag{D.5}$$

Using this, we derive the following inequality:

$$\mathbb{E}_\mathcal{S}[|\hat{F}(\theta^{(t)}) - F(\theta^{(t)})|] \leq \mathbb{E}_\mathcal{S}[|(\hat{F}(\theta^{(t)}) - \hat{F}(\phi)|] + \mathbb{E}_\mathcal{S}[|F(\theta^{(t)}) - F(\phi)|] + \mathbb{E}_\mathcal{S}[|\hat{F}(\phi) - F(\phi)|]$$

$$\leq 2L\varepsilon + \mathbb{E}_\mathcal{S}[|\hat{F}(\phi) - F(\phi)|]$$

$$\leq 2L\varepsilon + \mathbb{E}_\mathcal{S}\left[\left|\frac{1}{n}\sum_{i \in [n] \setminus \mathcal{I}}(\hat{f}(\phi; z_i) - F(\phi))\right|\right] + \mathbb{E}_{z_i : i \in \mathcal{I}}\left[\left|\frac{1}{n}\sum_{i \in \mathcal{I}}(\hat{f}(\phi; z_i) - F(\phi))\right|\right]$$

$$\leq 2L\varepsilon + \mathbb{E}_\mathcal{S}\left[\left|\frac{1}{n}\sum_{i \in [n] \setminus \mathcal{I}}(\hat{f}(\phi; z_i) - F(\phi))\right|\right] + \frac{BT}{n}$$

$$\leq \frac{BT+1}{n} + \frac{|[n] \setminus \mathcal{I}|}{n}\mathbb{E}_{z_i : i \in [n] \setminus \mathcal{I}}\left[\left|\frac{1}{|[n] \setminus \mathcal{I}|}\sum_{i \in [n] \setminus \mathcal{I}}(\hat{f}(\phi; z_i) - F(\phi))\right|\right]$$

$$\leq \frac{BT+1}{n} + \frac{|[n] \setminus \mathcal{I}|}{n}\left(CB\sqrt{\frac{1}{|[n] \setminus \mathcal{I}|}}\right)$$

$$\leq \frac{BT+1}{n} + CB\sqrt{\frac{1}{n}}.$$

The first and second inequality follows from the triangle inequality and (D.5), respectively. The third inequality is again from the triangle inequality while Assumption 2 gives us the fourth inequality. The sixth inequality is from Proposition 2.5.2 and Exercise 2.5.10 in [Ver18] where the last inequality naturally follows. Since the above bound holds for any $\theta^{(0)} \in \Theta$ and $\mathcal{I}_t$, this completes the proof of Corollary D.1

# E  Comparison with existing bounds

Table 1: Summary of generalization bounds for constant step-size SGD. In the column "Assumptions", Lip. assumes $L'$-Lipschitz $f(\,\cdot\,;z)$, Weak Lip. assumes Assumption 1, Bdd. assumes $f(\Theta;\mathcal{Z})\subset[0,B']$, and Bdd. Dev. assumes Assumption 2. If LHS of a bound does not contain expectation, that bound is a high-probability bound. For simplicity, we hide values other than $d,t,T,L,L',B,B',P,\xi,n$ in $\lesssim$ where $T=\log(LRn)$ and $\Theta\subset\mathcal{B}_R^d(0)$.

| Reference | Objective | Assumptions | Stable as $t\to\infty$ | Bound: $\Delta^{(t)}=\hat{F}(\theta^{(t)})-F(\theta^{(t)})$ |
|---|---|---|---|---|
| [HRS15] | Strongly convex & smooth | Lip. | ✓ | $\lvert\mathbb{E}[\Delta^{(t)}]\rvert\lesssim(L')^2/n$ [*] |
| Thm D.1 | | Weak Lip., & Bdd. Dev. | | $\lvert\mathbb{E}[\Delta^{(t)}]\rvert\lesssim BT/n$ [‖] |
| Thm D.2 | | | | $\mathbb{E}[\sup\lvert\Delta^{(t)}\rvert]\lesssim B\sqrt{T\log n/n}$ [‡‖] |
| Cor D.1 | | | | $\mathbb{E}[\lvert\Delta^{(t)}\rvert]\lesssim B\sqrt{1/n}$ [‖] |
| Thm 2.1 | | | | $\sup\lvert\Delta^{(t)}\rvert\lesssim B\sqrt{T\log n/n}$ [‡] |
| Cor 2.1–2.2 | | | | $\lvert\Delta^{(t)}\rvert\lesssim B\sqrt{1/n}$ |
| [HRS15] | Convex & smooth | Lip. | ✗ | $\lvert\mathbb{E}[\Delta^{(t)}]\rvert\lesssim(L')^2t/n$ [*] |
| [FV19] | | Lip. & Bdd. | | $\lvert\Delta^{(t)}\rvert\lesssim t\log^2 n/n+\sqrt{1/n}$ [¶] |
| [BFGT20] | Convex & non-smooth | Lip. | | $\lvert\mathbb{E}[\Delta^{(t)}]\rvert\lesssim(L')^2(\sqrt{t}+t/n)$ [*] |
| | | Lip. & Bdd. | | $\lvert\Delta^{(t)}\rvert\lesssim(L'\log n)^2(\sqrt{t}+t/n)+B'\sqrt{1/n}$ |
| [HRS15] | Non-convex & smooth | Lip. | | $\lvert\mathbb{E}[\Delta^{(t)}]\rvert\lesssim(L')^{\frac{2}{\beta c+1}}t^{\frac{\beta c}{\beta c+1}}/n$ [*§] |
| Thm 2.2 | Approx. piecewise strongly convex & smooth | Weak Lip. & Bdd. Dev. | ✓ | $\sup\lvert\Delta^{(t)}\rvert\lesssim B\sqrt{dT^2\log n/n}$ [‡] |
| | | | | $\sup\lvert\Delta^{(t)}\rvert\lesssim B\sqrt{T\log(nP)/n}+L\xi$ [‡] |

Table 2: Summary of generalization bounds for a (piecewise) contractive optimizer with update functions $g_1,\ldots,g_n$. The column "Non-cvx SGD" evaluates if a bound can be used for SGD on non-convex objectives without diverging as $t\to\infty$.

| Reference | Optimizer | Assumptions | Non-cvx SGD | Bound: $\Delta^{(t)}=\hat{F}(\theta^{(t)})-F(\theta^{(t)})$ |
|---|---|---|---|---|
| [KWS22] | Contractive $\frac{1}{n}\sum_{i=1}^{n}g_i$ | Lip. & $\lVert g_i(\Theta)\rVert\leq K$ | ✗ | $\lvert\mathbb{E}[\Delta^{(t)}]\rvert\lesssim KL'/n$ [*] |
| Thm D.1 | Contractive $g_i$ | Weak Lip. & Bdd. Dev. | ✗ | $\lvert\mathbb{E}[\Delta^{(t)}]\rvert\lesssim BT/n$ [‖] |
| Thm D.2 | | | | $\mathbb{E}[\sup\lvert\Delta^{(t)}\rvert]\lesssim B\sqrt{T\log n/n}$ [‡‖] |
| Cor D.1 | | | | $\mathbb{E}[\lvert\Delta^{(t)}\rvert]\lesssim B\sqrt{1/n}$ [‖] |
| Thm 4.1 | | | | $\sup\lvert\Delta^{(t)}\rvert\lesssim B\sqrt{T\log n/n}$ [‡] |
| Cor 2.1–2.2 [♭] | | | | $\lvert\Delta^{(t)}\rvert\lesssim B\sqrt{1/n}$ |
| Thm 4.1 | Approx. piecewise contractive $g_i$ | Weak Lip. & Bdd. Dev. | ✓ | $\sup\lvert\Delta^{(t)}\rvert\lesssim B\sqrt{T\log(nP)/n}+L\xi$ [‡] |

[*] The expectation is taken over $\{z_1,\ldots,z_n\},\{i_1,\ldots,i_t\},\theta^{(0)}$.
[‖] The expectation is taken over $\{z_1,\ldots,z_n\}$.
[‡] The supremum is taken over $\theta^{(t)}\in\bigcup_{\theta^{(0)}\in\Theta}\Psi_{\geq T}(\theta^{(0)})$ (see Definition 1 and (D.2)).
[§] $\beta$ denotes the smoothness parameter and the adaptive learning rate must satisfy $\eta_t\leq c/t$.
[¶] This bound is under 1-Lipschitz continuity of $f(\,\cdot\,;z)$ and $f(\Theta,\mathcal{Z})\subset[0,1]$.
[♭] Although Corollaries 2.1–2.2 are for SGD, the same result holds for contractive $g_i$ with the same proof.

# F Lower bound on uniform stability-based bound for piecewise strongly convex and smooth objectives

In this section, we show that the uniform stability-based bound [HRS15] is $\Omega(1)$ for piecewise strongly convex and smooth functions after sufficiently many SGD iterations in general. To this end, we first introduce the formal definition of the uniform stability and a standard tool for showing that the stability implies generalization, which are from Definition 2.1 and Theorem 2.2 in [HRS15].

**Definition 7** (Uniform stability). *A randomized algorithm $\rho$ is "$\varepsilon$-uniformly stable" if for all data sets $\mathcal{S}, \mathcal{S}' \in \mathcal{Z}^n$ such that $\mathcal{S}$ and $\mathcal{S}'$ differ in at most one example, we have*

$$\sup_{z \in \mathcal{Z}} \mathbb{E}_\rho[f(\rho(\mathcal{S}); z) - f(\rho(\mathcal{S}'); z)] \leq \varepsilon.$$

**Theorem F.1.** *If $\rho$ is $\varepsilon$-uniformly stable, then*

$$|\mathbb{E}_{\mathcal{S}, \rho}[\hat{F}(\rho(\mathcal{S})) - F(\rho(\mathcal{S}))]| \leq \varepsilon.$$

For the remaining section, we provide an example that

$$\sup_{z \in \mathcal{Z}} \mathbb{E}_\rho[f(\rho(\mathcal{S}); z) - f(\rho(\mathcal{S}'); z)] = \Omega(1),$$

regardless of $n$ where $\rho$ denotes sufficiently many SGD updates. Namely, the uniform stability-based bound based on Theorem F.1 is $\Omega(1)$.

Let $\Theta = [0, 4]$, $\mathcal{Z} = \{0, 1\}$, $\theta^{(0)} \sim \text{Unif}([0, 4])$, $\mathbb{P}(z = 0) = \mathbb{P}(z = 1) = \frac{1}{2}$, $f(\cdot\,; 0) = \min\{(x - 1)^2, \frac{1}{2} + \frac{1}{2}(x - 3)^2\}$, $f(\cdot\,; 1) = (x - 1)^2$, and the auxiliary gradient $\nabla f(2; 0) = 2$ at the non-differentiable point $\theta = 2$, i.e., $f(\cdot\,; z)$ is piecewise 1-strongly convex and 2-smooth with the partition $\mathcal{P} = \{[0, 2], (2, 4]\}$. For simplicity, choose $\eta = 1/3$ which can be generalized to arbitrary $\eta \in (0, 1)$. Under this setup, using Theorem 2.2, one can easily derive a generalization bound that does not increase with the number of SGD iterations and converges to zero as $n$ grows.

However, under the same setup and sufficiently many SGD iterations, the uniform stability-based bound is lower bounded by a constant regardless of $n$. To see this, let $\mathcal{S} = (0, \ldots, 0)$ and $\mathcal{S}' = (1, 0, \cdots, 0)$. Then, one can observe that $f([0, 2]; 0) \subset [0, 2]$ and $f((2, 4]; 0) \subset (2, 4]$, i.e. SGD iterates for $\mathcal{S}$ converge to either 1 or 3 depending on whether $\theta^{(0)} \in [0, 2]$ or $\theta^{(0)} \in (2, 4]$. Since we assumed $\theta^{(0)} \sim \text{Unif}([0, 4])$, we have

$$\lim_{t \to \infty} \mathbb{E}_{\theta^{(t)}}[f(\theta^{(t)}; 1)] = \frac{1}{2}f(1; 1) + \frac{1}{2}f(3; 1) = 2 \tag{F.1}$$

where $\theta^{(t)}$ denotes a random parameter generated by $t$ SGD updates for $\mathcal{S}$, from $\theta^{(0)} \sim \text{Unif}([0, 4])$. Furthermore, we have $f(\Theta; 1) \subset [0, 2]$ and $f([0, 2]; 0) \subset [0, 2]$. This implies that if a single SGD update for $\mathcal{S}'$ use the first sample in $\mathcal{S}'$ (i.e. $z = 1$), which occurs with high probability under sufficiently many SGD iterations, then the SGD iterates will converge to 1 almost surely. In other words, we have

$$\lim_{t \to \infty} \mathbb{E}_{\phi^{(t)}}[f(\phi^{(t)}; 1)] = f(1; 1) = 0 \tag{F.2}$$

where $\phi^{(t)}$ denotes a random parameter generated by $t$ SGD updates for $\mathcal{S}'$, from $\phi^{(0)} \sim \text{Unif}([0, 4])$. Combining (F.1) and (F.2) implies a constant lower bound on the uniform stability-based bound, regardless of $n$, under sufficiently many SGD iterations. We note that the same conclusion can also be derived for any $\eta \in (0, 1)$ as long as the number of SGD iterations is large enough.