# OpenReview forum: "Generalization Bounds for Stochastic Gradient Descent via Localized $\varepsilon$-Covers"
_NeurIPS.cc/2022/Conference — NeurIPS 2022 Accept_

### Official Review · Reviewer_2vrF · 2022-06-15

**Rating:** 8
**Confidence:** 3
**Soundness:** 3 good
**Presentation:** 3 good
**Contribution:** 2 fair

**Summary:**

The authors study the generalization properties of parametric models trained with SGD. While classical statistical analysis of the generalization error $F(\hat{\theta}) - \hat{F}(\hat{\theta})$ rely on uniform bounding on the parameter space $\Theta$, thus ignoring the nature of the optimization process and leading to rates of the form $\sqrt{d/n}$ where $d$ is the dimension of $\Theta$ and $n$ is the number of samples, the authors propose to bound instead the process $F(\hat{\theta}) - \hat{F}(\hat{\theta})$ over points that are reached under the action of the SGD on the parameter space $\Theta$.
Under a contraction hypothesis of the updates, two different points with synchronous SGD updates will eventually be close. In particular, the action of the SGD on $\Theta$ can be reduced to the action of the SGD starting from the point 0 (see Lemma 2.1) with $T$ iterations. Since the number of SGD trajectories starting from 0 with $T$ iterations does not depend on the dimension of the parameter space (coarsely upper-bounded by $n^T$), the dependency in $d$ in the generalization bound is removed.
The authors prove these results in the strongly convex and smooth case and in the more general case of perturbated piecewise smooth strongly convex functions. In particular, they highlight the fact that the latter can encompass standard generalization bound as any piecewise smooth function can be approximated by a piecewise smooth strongly convex function at the cost of increasing the number of pieces thus recovering the standard dimension dependent generalization bounds (see Remark, l. 196- 201).
Yet they showcase on several learning paradigms that they can improve this worst case bound and surpass state-of-the-art bounds.

**Questions:**

See Weaknesses.


**Limitations:**

The authors clearly show that their result is only a part of the equation when it comes to the question of the risk excess bound. Indeed, they claim they do not control the term $\hat{F}( \hat{\theta}) - \hat{F}( \theta^*)$ where $\theta^*$ is the true minimum $\inf_\theta F(\theta)$.

**Strengths And Weaknesses:**

Strengths:
- The idea of studying only the parameter space under the action of SGD is super natural and elegant.
- The paper is well-written and easy to follow even for non-experts, arguments can be easily understood: indeed, under the contraction property, it suffices to study the trajectories starting from 0 with a finite number of iterations T.
- The framework is broad and as shown in their remark, it covers almost all cases of standard ML problems (even though it does not improve known results).
- In Section 3, the authors do a good job at convincing that their results can improve generalization bounds in many ML settings.
- The comparison to other works in related fields seems to be correctly made and from the samples I've read, the approach of the authors do seem quite different from other cited works yet it seems some results were already covered by [KSW22].

Weaknesses:
- Novelty of the results?  Even though the approach seems different and more intuitive, I believe the work of the authors is somehow a less general version [KWS22]. Indeed, the results are with high probability instead of in expectation and only piecewise contraction is required (instead of global contraction) yet it seems like a marginal improvement. In particular, their whole paragraph 2.1 is readily covered by [KSW22] and it should have been highlighted more clearly right in this section (instead of simply being mentioned in the Discussion section at the end of the paper).
- (Minor) A more intuitive proof than the one provided in the Appendix would have been to use a standard metric entropy reasoning: Lemma 2.1 provides an \epsilon covering of $\psi_{\geq T}(\theta_0)$ of cardinality $| \psi_T( 0 ) | \leq n^{T(\epsilon)}$, hence we can upper bound with a chaining argument $E |F(\hat{\theta}) - \hat{F}(\hat{\theta})| \leq \sqrt{\log(n) / n} \int_{0}^{O(1)}  T(\epsilon)^{0.5} d \epsilon$ and recover a result with high probability using a Bousquet concentration bound. Is it because of the suboptimal $1/\sqrt{n}$ dependance that this reasoning was not used? If it is the case, I believe it should have been quickly mentioned as it would have been the natural argument for a statistician.

---

> ### Author Response · Authors · 2022-08-02
> **Response to Reviewer 2vrF**
>
> We thank the reviewer for their valuable comments. We address the reviewer’s concerns point by point below, and kindly ask the reviewer to reassess our work in light of our response.
>
> **Novelty and comparison with [KWS22]:**
> Following the reviewer's suggestion, we added a paragraph in Section 2.1 in Lines 174-183, a comment in Lines 229-231, and Table 2 in Appendix E to compare our work with [KWS22] in the revised version of our paper.
>
> Contractivity of SGD (under the geodesic distance) has indeed been used to derive generalization bounds in the **concurrent** work [KWS22]. Although the localized covering construction in Lemma 2.1 relies on the same principle, our results differ from those in [KWS22] on two key aspects:
>
> - First and foremost, in Section 2.2, the localized covering argument used in Lemma 2.1 can also be applied to a wide range of non-convex problems in which the uniform stability-based argument used in [KWS22] provably does not hold (see Appendix F); thus, extending the results of [KWS22] to cover non-convex objectives is highly non-trivial.
> - Further, the generalization bounds in [KWS22] are provided in expectation, i.e. $|\mathbb E[\hat F(\theta^{(t)})-F(\theta^{(t)})]|\le O(1/n)$, whereas we provide high-probability bounds, i.e. $|\hat F(\theta^{(t)})-F(\theta^{(t)})| \le \widetilde O(1/\sqrt{n})$. When translated to a bound in expectation via standard tools, our results read
>  $|\mathbb E[\hat F(\theta^{(t)})-F(\theta^{(t)})]| \le \widetilde O(1/n)$ and $\mathbb E[|\hat F(\theta^{(t)})-F(\theta^{(t)})|] \le\widetilde O(1/\sqrt{n})$; see Appendices D and E for details.
>
> A more thorough comparison is provided in Table 2 in Appendix E in the revised paper, and a summary version is given below.
>
>
>
> **Table 2:** Summary of generalization bounds for stochastic (piecewise) contractive optimizers with update functions $g_1,\dots,g_n$. We hide all quantities other than $n,d,P,\xi$ and all polylogarithmic quantities other than $\log P$ using $\lesssim$. Here, the supremum is taken over all possible SGD initializations and all possible sequences of indices of length $\Omega(\log n)$ used for SGD updates.
> | Reference | Objective | Assumptions | Non-cvx SGD | Bound: $\Delta^{(t)}=\hat F(\theta^{(t)})-F(\theta^{(t)})$ |
> |:-------------:|:------------|:----------------|:------------------|:--------------------------------------------------------------------|
> | [KWS22] | Contractive $\frac1n\sum_ig_i$ | Lipschitz & $\|g_i\|\le K$|&#x2718; | $\lvert\mathbb E_{\text{samples,SGD}}[\Delta^{(t)}]\rvert\lesssim1/n$ |
> | Thm D.1 | Contractive $g_i$ | Weak Lipschitz & Bounded deviation |&#x2718; | $\lvert\mathbb E_{\text{samples}}[\Delta^{(t)}]\rvert\lesssim1/n$|
> | Thm D.2 | Contractive $g_i$ |Weak Lipschitz & Bounded deviation |&#x2718; | $\mathbb E_{\text{samples}}[\sup\lvert\Delta^{(t)}\rvert]\lesssim \sqrt{1/{n}}$ |
> | Thm 4.1 | Contractive $g_i$ | Weak Lipschitz & Bounded deviation |&#x2718; | $\sup\lvert\Delta^{(t)}\rvert\lesssim \sqrt{1/{n}}$|
> | Thm 4.1 | Approx. piecewise contractive $g_i$ |Weak Lipschitz & Bounded deviation |&#10004; | $\sup\lvert\Delta^{(t)}\rvert\lesssim\sqrt{\log P/n}+\xi$|
>
>
>
> We emphasize that uniform stability-based approaches for SGD (e.g. [HRS15, KWS22]) do not adapt to non-convexity in general. To our knowledge, the best stability-based bound for general non-convex objectives grows with the number of SGD iterations [HRS15] (see Table 1 in Appendix E in the revised paper), while existing non-diverging bounds require restrictive properties, e.g. global convergence induced by the Polyak-Lojasiewicz inequality [CP18], or a guarantee that the optimizer does not escape from a locally contractive region, e.g. see Section 4.1 in [KWS22]. In particular, as we mentioned before, the uniform stability approach used in [HRS15, KWS22] provably breaks down for piecewise strongly convex and smooth functions (as shown in Appendix F in the revised paper). In contrast, our localized covering technique enables us to derive a **non-diverging** generalization bound for finite perturbations of piecewise strongly convex and smooth functions (which can be extended to a piecewise contractive optimizer, cf. Theorem 4.1).
>
> **About applying chaining:**
> We thank the reviewer for the intriguing question. We note that our localized covering construction $\Psi_T(0)$ is sample dependent; thus, the covering sets indeed depend on the data, unlike in standard metric entropy reasoning. Consequently, directly applying chaining in this context is non-trivial; yet it is indeed an interesting future work.
>
> ---
> We would be happy to clarify any concerns or answer any questions that may come up during the discussion period.
>
> ___
> [CP18] “Stability and Generalization of Learning Algorithms that Converge to Global Optima,” ICML, 2018

---

> > ### Comment · Reviewer_2vrF · 2022-08-03
> > **Response author rebuttal**
> >
> > Dear authors,
> >
> > Thank you for clarifying my concerns. Indeed, after re thinking about it, the method goes well beyond contraction as it lets the optimization trajectories having multiple (yet finite and dimension independent) accumulation points provided updates are synchronised ; I believe it actually lets the optimizer being locally contractant.
> >
> > I raise my score to 8 and congratulate the authors for their nice work.

---

> > > ### Author Response · Authors · 2022-08-03
> > > **Thank you for the prompt response**
> > >
> > > We sincerely appreciate the reviewer's reevaluation. We reiterate that we would be happy to clarify any further questions that may come up during the discussion period.

---

### Official Review · Reviewer_PjwR · 2022-07-07

**Rating:** 7
**Confidence:** 3
**Soundness:** 3 good
**Presentation:** 4 excellent
**Contribution:** 4 excellent

**Summary:**

The paper studies the generalization gap for objectives that are optimized
via stochastic gradient descent (SGD). By leveraging the fact that SGD
updates are contractive when all component objective functions are strongly
convex, the authors are able to apply an $\epsilon$-covering argument that
removes the dependence of d in the generalization gap for many
situations. Moreover, by arguing that any smooth, bounded non-convex
function is within an $\epsilon$ (in the $L^{\infty}$ norm) of piecewise strongly
convex functions, they are able to generalize their argument to a larger
share of non-convex objectives. Namely, they are able to demonstrate an
improvement in the generalization gap in the class of multi-index models and
K-means with hard label assignment.

**Questions:**

[Q1] Are the best known bounds stated for K-means (soft and hard) and
multi-index models that were beat in this paper for the tail probabilities
or for the average SGD iterates?

[Q2] Is there any general theory describing when a smooth function can be
represented in the form of described in Proposition 2.1 with very few
piecewise components?

**Limitations:**

No, the authors have not addressed the societal impact if their work is misconstrued or incorrect.

**Strengths And Weaknesses:**

Overall, I found this to be a well-written and well thought out paper.

[Originality]

Strengths:
The core of the paper seems to rely on two different insights:
+ The set of all possible SGD iterates at step T form an $\epsilon$-covering for
  all future SGD iterates with an appropriately chosen T
+ Any smooth, bounded non-convex function can be arbitrarily close to a set of
  piecewise smooth strongly convex functions (with the number of required
  functions possibly but not always growing exponentially in the ambient dimension d).

Armed with these two observations, the authors are able to argue an upper
bound on the generalization gap that does not necessarily rely on the
dimension. The combination of these two ideas permits the authors to
demonstrate an improvement in some common ML models (K-means and
multi-index models).

Weaknesses:
The paper mentioned that there was some previous inspiration based on
understanding the Hausdorff dimension of an SGD iterate chain, so in that
sense, the paper had some previous related ideas. However, I found this
connection to build more understanding rather than diminish the work done in
this paper.

[Quality]

Strengths:
While the theoretical framework in Theorem 2.1 and 2.2 are of independently
of interest, the application of these theorems to some real models is
powerful. This demonstrates that this perspective of using an
algorithmic-dependent stability can provide some benefit on top of general
methods that only consider the objective function and not the optimization
routine.

Weaknesses:
I do not have any notable complaints with the work presented.


[Clarity]

Strengths:
The paper was very clear and easy to follow (with a couple exceptions, see
detailed comments below).

Weaknesses:
NA

[Significance]

Strengths:
The generalization gap is a widely studied object in modern day ML. The
techniques used here could spark other ideas on how to use computational
geometry to better understand the nature of SGD in deep learning.

Weaknesses:
I could not tell from reading the paper whether the fact that these results
were upper bounds on the tail probabilities made this easier than the
expectation bounds mentioned in paper. I am not certain if this could be
drawn out a bit more, but it would be helpful for the reader to understand.

---

> ### Author Response · Authors · 2022-08-02
> **Response to Reviewer PjwR**
>
> We thank the reviewer for their positive feedback and valuable comments. We provide our point by point response below.
>
> **Bounds in expectation:**
> Even though our main focus was high-probability bounds, the localized covering argument can also be used for deriving generalization bounds in expectation. To demonstrate this, we additionally derived bounds in expectation (Theorems D.1, D.2, and Corollary D.1 in the revised paper) and compared them with existing bounds in Appendix D in the revised paper.
>
> **Best known bounds for K-means (soft and hard) and multi-index models:**
> Existing generalization bounds for K-means clustering and multi-index linear models in the literature are not specialized for SGD; they are in general based on uniform convergence (i.e. algorithm-independent). An algorithm-dependent exception is given for the variants of Lloyd’s algorithm for $K$-means clustering. While most of these bounds are for the tail probability, some of them are for the expected generalization gap, e.g. [Zha04, Lev13]. We note that the bound for the (average) SGD iterates can be (crudely) derived as a special case of existing algorithm-independent bounds.
>
> **Theory for representing a smooth function with few piecewise components:**
> We thank the reviewer for the intriguing question. We are not aware of any general properties of a function that a small number of pieces suffice for the approximation. However, from our applications in Section 3, we expect that if an objective function involves certain symmetries, then a small number of pieces should be sufficient. For example, in the K-means clustering under the soft label setup, each objective is a function of the 2-norm, i.e. symmetric with respect to some point, say $x$. Based on this property, we approximate the objective using quadratic functions centered around $x$, which takes a dimension-free number of pieces. In light of this observation, if the objective is a (piecewise) function of the 2-norm, then one may derive a dimension-independent bound using Theorem 2.2. We think that extending this idea and exploring more applications of our approach are interesting future directions.
>
> ___
>
> We would be happy to clarify any concerns or answer any questions that may come up during the discussion period.

---

### Official Review · Reviewer_pTmX · 2022-07-09

**Rating:** 3
**Confidence:** 4
**Soundness:** 1 poor
**Presentation:** 2 fair
**Contribution:** 2 fair

**Summary:**

This submission considers a localized covering technique to analyze the generalization bounds of SGD. The technique claims to construct an $\epsilon$-net which is spanned by the points on the trajectory reachable by SGD iterations, which leads to the dimension-independent bounds with constant step size and without using early stopping.

**Questions:**

--> Assumption 2: I do find the bounded deviation assumption to be quite strong. Is it possible to remove this assumption?

--> Equation (1.3): I am a little surprised that the generalization bound is independent from the iteration count $t$. What is the rationale behind this claim?

--> Line 213: "multi-index linear models" why is this type of models fundamentally different from "single-index models"?
--> Equation (1.4): have authors considered the SGD with minibatch?

--> In lemma 2.1, the authors have implicitly assumed the boundedness of the iteration sequence $\theta^{(t)}$. Is this assumption reasonable?

--> Main concern: I find the proof to lemma 2.1 to be dubious. In particular, the claim $\|\theta^{(t)} - \phi\|\le \gamma^T R:= \varepsilon$ does not seem correct.

--> Main concern: I am also surprised that there is no expectation involved in the statement of lemma 2.1.

--> Line 272: I doubt about the correctness of applying results with respect to differentiable functions on non-differential objective functions.

**Limitations:**

Strong assumptions made in the paper and unclear proof makes this submission require much more careful scrutiny.

**Strengths And Weaknesses:**

- Strength: the narrative in general flowing well and the paper has a clear structure

- Weakness:

--> Assumptions made in the paper are strong.

--> Key lemma needs further explanation.

---

> ### Author Response · Authors · 2022-08-02
> **Response to Reviewer pTmX**
>
> We thank the reviewer for their valuable comments. We address the reviewer’s concerns point by point below, and kindly ask the reviewer to reassess our work in light of our response.
>
> **Assumption 2 is strong. Is it possible to remove?:**
> In the context of our paper, the bounded deviation is a rather **standard** assumption for deriving high-probability generalization bounds. In fact, existing works often make the even stronger assumption that the objective is uniformly bounded ($f(\theta,z)\in[0,B]$ for all $\theta$ and $z$), see e.g.  [SSSSS09, FV19, BFGT20]. In contrast, Assumption 2 allows for regularizers without an additional cost in our generalization bounds.
>
> There has been progress to relax this condition, e.g. by assuming bounded tail deviation [CDE+21]; however, incorporating their technique with our localized cover approach is highly non-trivial, and left as future work.
>
> **Main concern I: Proof to Lemma 2.1 is dubious:**
> We have improved the proof of Lemma 2.1 and fixed a typo that was the cause of suspicion. The statement $\|\theta^{(t)}-\phi\|\le\gamma^T R=:\varepsilon$ was written for $T_\varepsilon=\frac{\log(R/\varepsilon)}{\log(1/\gamma)}$ (without the ceiling) and should instead be written as $\|\theta^{(t)}-\phi\|\le\gamma^T R\le\varepsilon$ for $T_\varepsilon=\left\lceil\frac{\log(R/\varepsilon)}{\log(1/\gamma)}\right\rceil$ (with the ceiling). This is fixed in the revised paper.
>
>
> **Main concern II: Surprised to see no expectation in Lemma 2.1:**
> The sets $\Psi_T$ and $\Psi_{\geq T}$ in Lemma 2.1 do not involve any randomness incurred by the SGD algorithm; they describe **all feasible points** that can be generated by SGD trajectories. For a given data, the union of all feasible SGD trajectories is a deterministic set.
>
> In the updated version, this is clarified in Line 104.
>
> **Rationale behind Equation (1.3) being independent of iteration count:**
> At a high level, after sufficiently many iterations of SGD, the parameters converge and get trapped in certain regions. Since our technique identifies these regions and considers the worst-case scenario within these sets, taking more iterations will not change the generalization bound.
>
> More specifically, as discussed in Section 2.2, we show that if objectives are finite perturbations of piecewise strongly convex and smooth functions, then all SGD iterates **after sufficiently many iterations** can be contained in finitely many balls, whose number is independent of the iteration count $t$ (see Lemma 2.1 for the strongly convex and smooth case). Under this observation and by applying a slight modification of the standard covering argument (see Theorem B.1 in Appendix), we could derive Equation (1.3), independent of the iteration count $t$.
>
> **Multi-index vs single-index models:**
> Multi-index and single-index models are both fundamental concepts in statistical learning, see e.g. [LD89, Li91, CHM11, BK19, GMM+20].
>
> Multi-index linear models are generalizations of single-index models, especially in terms of their expressive power. In particular, while a single-index model only considers a non-linear function of a single linear component of an input (e.g. support vector machines and logistic regression for binary classification), a multi-index model can encode multiple (i.e. $K$ many) linear components of the input at once so that it can express more complex functions (e.g. wide neural networks, multi-class support vector machines, and logistic regression, etc.).
>
> **SGD with minibatch:**
> Indeed, our techniques can be easily extended to mini-batch SGD; yet, we only considered SGD in this current paper for simplicity. In Lines 331-334 (“Extension to mini-batch setup …” paragraph), we provided a discussion on how to extend our localized covering technique to the mini-batch setup.
>
> **Implicitly assumed boundedness of iterations:**
> Note that we consider the projected SGD throughout the paper (Equation (1.4)); thus, all the iterates are bounded. In the context of our work, the projection step is **common**, see e.g. [SSSSS09, HRS15, FV19]. Nevertheless, under the presence of an $\ell_2$-regularizer and the Lipschitz continuity of the objective, the projection step can be removed, as discussed in Section 4.
>
> In the updated version, this is clarified in Line 109.

---

> > ### Author Response · Authors · 2022-08-02
> > **Response to Reviewer pTmX**
> >
> > **Applying results with differentiable functions on non-differential objectives:**
> > We would like to note that Theorem 3.3 is not a corollary of Theorem 2.2, and its proof is different.
> > Nevertheless, as highlighted in Lines 175-176 in the initial submission, we use SGD with **auxiliary gradients** (as defined in Definition 2) when differentiability is an issue. As such, Theorem 2.2 can also be applied for the non-differentiable objectives as long as auxiliary gradients are used in SGD updates satisfying the conditions in Theorem 2.2.
> >
> > We clarified this point in Lines 220-221 of the revised paper.
> >
> > ---
> > We would be happy to clarify any concerns or answer any questions that may come up during the discussion period.
> >
> > ---
> >
> > - [LD89] “Regression analysis under link violation,” The Annals of Statistics, 1989
> > - [Li91] “Sliced inverse regression for dimension reduction,” Journal of the American Statistical Association, 1991
> > - [CHM11] “Single and multiple index functional regression models with nonparametric link,” The Annals of Statistics, 2011
> > - [BK19] “On deep learning as a remedy for the curse of dimensionality in nonparametric regression,” The Annals of Statistics, 2019
> > - [GMM+20] “When Do Neural Networks Outperform Kernel Methods?,” NeurIPS, 2020

---

### Official Review · Reviewer_FYvm · 2022-07-11

**Rating:** 7
**Confidence:** 3
**Soundness:** 4 excellent
**Presentation:** 4 excellent
**Contribution:** 3 good

**Summary:**

The paper proposes a new covering technique localized for SGD trajectories. For objectives that are perturbations of piecewise strongly convex and smooth functions, the proposed approach provides a dimension-independent upper bound on the generalization error. The authors further show that the approach gives rise to improved rates for several important applications, including multi-class SVMs and k-means clustering.

**Questions:**

See above.

**Limitations:**

Limitations and possible extensions are sufficiently assessed in the discussion section.

**Strengths And Weaknesses:**

The paper is very well written; theoretical results are clearly presented and motivated. The proof sketches are instructive and give a good preview of the full arguments in the supplemental. The part of the theoretical results that I checked seems correct. It might be helpful for the reader to provide a table with the best-known rates and assumptions/ requirements in the previous and concurrent literature -- in the current version this is provided only in-text (e.g., Remark on page 4 and Discussion section), which is harder to read.

The paper is not in my direct area of research, so it is difficult to judge novelty and strength. Based on the background given in the paper, the results improve over the SOTA for several instances of the problem class considered, which would be a strong contribution.

---

> ### Author Response · Authors · 2022-08-02
> **Response to Reviewer FYvm**
>
> We thank the reviewer for their positive evaluation and thoughtful feedback.
>
> Following the reviewer’s suggestion, we added two tables (Tables 1 & 2) in Appendix E of the updated manuscript. These tables will be included in the camera-ready version of our paper. We provide the summary versions below (see Appendix E for the full versions).
>
> **Table 1:** Summary of generalization bounds for constant step-size SGD. We hide all quantities other than $n,d,P,\xi$ and all polylogarithmic quantities other than $\log P$ using $\lesssim$. Here, the supremum is taken over all possible SGD initializations and all possible sequences of indices of length $\Omega(\log n)$ used for SGD updates.  $\beta$ denotes the smoothness parameter, and $c$ is some constant associated with the step size.
> | Reference | Objective | Assumptions | Stable as $t\rightarrow\infty$ | Bound: $\Delta^{(t)}=\hat F(\theta^{(t)})-F(\theta^{(t)})$ |
> |:-------------:|:------------|:----------------|:------------------|:--------------------------------------------------------------------|
> | [HRS15] | Str. conv. & smooth | Lipschitz |&#10004; | $\lvert\mathbb E_{\text{samples,SGD}}[\Delta^{(t)}]\rvert\lesssim1/n$ |
> | Thm D.1 | Str. conv. & smooth | Weak Lipschitz & Bounded deviation |&#10004; | $\lvert\mathbb E_{\text{samples}}[\Delta^{(t)}]\rvert\lesssim1/n$|
> | Thm D.2 | Str. conv. & smooth |Weak Lipschitz & Bounded deviation |&#10004; | $\mathbb E_{\text{samples}}[\sup\lvert\Delta^{(t)}\rvert]\lesssim \sqrt{1/{n}}$ |
> | Thm 2.1 | Str. conv. & smooth | Weak Lipschitz & Bounded deviation |&#10004; | $\sup\lvert\Delta^{(t)}\rvert\lesssim \sqrt{1/{n}}$|
> | [HRS15] | Conv. & smooth |Lipschitz |&#x2718; | $\lvert\mathbb E_{\text{samples,SGD}}[\Delta^{(t)}]\rvert\lesssim t/n$|
> | [FV19] | Conv. & smooth |Lipschitz & Bounded |&#x2718; | $\lvert\Delta^{(t)}\rvert\lesssim t/n+\sqrt{1/n}$|
> | [BFGT20] | Conv. & non-smooth |Lipschitz |&#x2718; | $\lvert\mathbb E_{\text{samples,SGD}}[\Delta^{(t)}]\rvert\lesssim \sqrt{t}+t/n$|
> | [BFGT20] | Conv. & non-smooth |Lipschitz & Bounded |&#x2718; | $\lvert\Delta^{(t)}\rvert\lesssim \sqrt{t}+t/n+\sqrt{1/n}$|
> | [HRS15] | Non-conv. & smooth |Lipschitz |&#x2718; | $\lvert\mathbb E_{\text{samples,SGD}}[\Delta^{(t)}]\rvert\lesssim t^{\beta c/(\beta c+1)}/n$|
> | Thm 2.2 | Non-conv. & smooth |Weak Lipschitz & Bounded deviation |&#10004;| $\sup\lvert\Delta^{(t)}\rvert\lesssim\sqrt{d/n}$|
> | Thm 2.2 | Approx. piecewise str. conv. & smooth |Weak Lipschitz & Bounded deviation |&#10004; | $\sup\lvert\Delta^{(t)}\rvert\lesssim\sqrt{\log P/n}+\xi$|
>
> **Table 2:** Summary of generalization bounds for stochastic (piecewise) contractive optimizers with update functions $g_1,\dots,g_n$.
> | Reference | Objective | Assumptions | Non-conv. SGD | Bound: $\Delta^{(t)}=\hat F(\theta^{(t)})-F(\theta^{(t)})$ |
> |:-------------:|:------------|:----------------|:------------------|:--------------------------------------------------------------------|
> | [KWS22] | Contractive $\frac1n\sum_ig_i$ | Lipschitz & $\|g_i\|\le K$|&#x2718; | $\lvert\mathbb E_{\text{samples,SGD}}[\Delta^{(t)}]\rvert\lesssim1/n$ |
> | Thm D.1 | Contractive $g_i$ | Weak Lipschitz & Bounded deviation |&#x2718; | $\lvert\mathbb E_{\text{samples}}[\Delta^{(t)}]\rvert\lesssim1/n$|
> | Thm D.2 | Contractive $g_i$ |Weak Lipschitz & Bounded deviation |&#x2718; | $\mathbb E_{\text{samples}}[\sup\lvert\Delta^{(t)}\rvert]\lesssim \sqrt{1/{n}}$ |
> | Thm 4.1 | Contractive $g_i$ | Weak Lipschitz & Bounded deviation |&#x2718; | $\sup\lvert\Delta^{(t)}\rvert\lesssim \sqrt{1/{n}}$|
> | Thm 4.1 | Approx. piecewise contractive $g_i$ |Weak Lipschitz & Bounded deviation |&#10004; | $\sup\lvert\Delta^{(t)}\rvert\lesssim\sqrt{\log P/n}+\xi$|
>
> ___
> We would be happy to clarify any concerns or answer any questions that may come up during the discussion period.

---

> > ### Comment · Reviewer_FYvm · 2022-08-09
> > **Response to author reply**
> >
> > Thanks to the authors for their reply. I am keeping my score of acceptance.

---

### Official Review · Reviewer_4jdi · 2022-07-15

**Rating:** 8
**Confidence:** 4
**Soundness:** 3 good
**Presentation:** 3 good
**Contribution:** 4 excellent

**Summary:**

The paper studies generalization gap for stochastic gradient descent. Since the generalization gap can be expressed in terms of the number of fixed-radius balls required to cover the domain on the function, it suffices to bound the number of balls covering the SGD trajectory. For strongly convex functions, the bound is derived based on the fact that the gradient descent step is a contractor, i.e. it decreases the distance between points by a constant factor. For a non-convex function, the paper uses the fact that its gradients can be approximated by gradients of a piecewise strongly convex function. While the number of pieces can depend exponentially on the dimension, providing no improvement in the worst case, the paper presents a number of applications that have non-trivial number of pieces, improving state-of-the-art for the respective problems and sometimes significantly improving dependence on dimension.


**Questions:**

I think it would be great to expand the derivation of the second term in the second line of Equation (B.3)


**Limitations:**

Yes

**Strengths And Weaknesses:**

The paper is easy to read and it clearly conveys its ideas. While I’m not sure that all mentioned applications are of great importance (in particular, single-layer neural networks), I believe that the presented applications shows the potential of this approach, especially in presence of regularization, as outlined in Discussion.

---

> ### Author Response · Authors · 2022-08-02
> **Response to Reviewer 4jdi**
>
> We thank the reviewer for their positive evaluation and thoughtful feedback.
>
>
> Following the reviewer’s suggestion, we made the following changes in the updated version of our paper:
>
> - We expanded the derivation of the second line of Equation (B.3) in order to clearly demonstrate that each term in the bound can be controlled separately.
> - We added more explanation about this derivation in Lines 579-582 in the revised manuscript.
>
> ___
> We would be happy to clarify any concerns or answer any questions that may come up during the discussion period.

---

### Author Response · Authors · 2022-08-02
**Summary of the revision**

Dear Reviewers and Area Chair,

We deeply appreciate your continuing time and effort to provide detailed comments on our paper. To best respond to your comments, we revised our paper with additional clarifying content. Below is a short summary of the significant updates (all updates are color coded in the revision).

- We added two tables summarizing existing work and our results in Appendix E (Reviewer FYvm).
- Theorem 4.1 is updated to cover approximate piecewise contractive optimizers; previously, it only covered piecewise contractive optimizers.
- We included additional generalization bounds in expectation in Appendix D (Reviewer PjwR).
- We added a paragraph discussing the work [KWS22] in Section 2.1 (Reviewer 2vrF).
- We showed that the uniform stability approach breaks down for piecewise strongly convex and smooth functions in Appendix F (Reviewer 2vrF).
- We clarified the derivation of Equation (B.3) (Reviewer 4jdi).
- We improved the proof of Lemma 2.1 (Reviewer pTmX).


Please refer to our detailed response to each reviewer where we address all individual comments with pointers to the corresponding update in the revised paper.

Sincerely,

---

### Meta-Review · Area_Chair_zvGt · 2022-08-25

**Recommendation:** Accept
**Confidence:** Certain

**Metareview:**

This paper introduces a new framework for proving generalization bounds for SGD that is based on covering the space of trajectories. When the underlying function is smooth and strongly convex, the fact that gradient descent contracts the distance between points by a constant factor, can be used to construct a good cover. More interestingly, their framework can be applied to nonconvex functions too, by approximating them by a piecewise strongly convex function. In general, the number of pieces will grow exponentially with the dimension, but in some important applications, like multiclass SVMs and k-means clustering, they are able to use their framework to derive interesting new generalization bounds. The paper is well-written and there was almost uniform consensus that it ought to be accepted.

**Award:**

No

---

### Decision · Program_Chairs · 2022-09-14

Accept